# ReinFlow: Fine-tuning Flow Matching Policy with Online Reinforcement Learning

**Tonghe Zhang**
Robotics Institute
Carnegie Mellon University
tonghez@andrew.cmu.edu

**Chao Yu**[*]
Shenzhen International Graduate School
Tsinghua University
zoeyuchao@gmail.com

**Sichang Su**
Department of Aerospace Engineering
The University of Texas at Austin
sichang_su@utexas.edu

**Yu Wang**
Department of Electronic Engineering
Tsinghua University
yu-wang@mail.tsinghua.edu.cn

## Abstract

We propose ReinFlow, a simple yet effective online reinforcement learning (RL) framework that fine-tunes a family of flow matching policies for continuous robotic control. Derived from rigorous RL theory, ReinFlow injects learnable noise into a flow policy's deterministic path, converting the flow into a discrete-time Markov Process for exact and straightforward likelihood computation. This conversion facilitates exploration and ensures training stability, enabling ReinFlow to fine-tune diverse flow model variants stably, including Rectified Flow [34] and Shortcut Models [18], particularly at very few or even one denoising step. We benchmark ReinFlow in representative locomotion and manipulation tasks, including long-horizon planning with visual input and sparse reward. The episode reward of Rectified Flow policies obtained an average net growth of 135.36% after fine-tuning in challenging legged locomotion tasks while saving denoising steps and 82.63% of wall time compared to state-of-the-art diffusion RL fine-tuning method DPPO [42]. The success rate of the Shortcut Model policies in state and visual manipulation tasks achieved an average net increase of 40.34% after fine-tuning with ReinFlow at four or even one denoising step, whose performance is comparable to fine-tuned DDIM policies while saving computation time for an average of 23.20% . Code, model, and checkpoints available on the project website: https://reinflow.github.io/

## 1 Introduction

Recent years have seen rapid progress in the training of robotic models via imitation learning. Flow matching models, which combine the trifecta of precise modeling, fast inference, and minimal implementation, have emerged as a robust alternative to diffusion policies and a popular choice for robot action generation [8, 6, 57].

However, due to the scarcity of robot data and the embodiment gap, flow policies and many other imitation learning policies are often trained from datasets with mixed quality [13], incurring suboptimal success rates even after supervised fine-tuning [6]. Although collecting more data may alleviate this problem, recent work [31] suggests that scaling the quantity of data may not be a panacea because, in a given environment, the success rate quickly plateaus when we only increase the number of demonstrations. Worse still, as an imitation learning method, flow policies lack a built-in exploration

---

[*]Corresponding author

39th Conference on Neural Information Processing Systems (NeurIPS 2025).

mechanism, implying that robots trained on imperfect data could struggle to complete challenging tasks where the agent needs to surpass expert demonstrations [5].

Online reinforcement learning (RL) offers a promising solution to these challenges. By learning through trial and error, RL promises to overcome limitations associated with imperfect expert data and even achieve superhuman performance. Although recent studies have shown that it is possible to fine-tune diffusion policies via RL [30, 42], training flow models with online RL remains technically challenging.

First, the theory has established that the refinement of a stochastic policy is vital for continuous control problems [49]. However, for conditional flows, whose sample path is governed by a neural ordinary differential equation (ODE), even the log probability-a key factor that measures this stochasticity-could be challenging to obtain and unstable to backpropagate through. This problem is exacerbated when we infer flow policies at very few denoising steps, where the discretization error becomes large at a low inference cost. Second, compared to offline RL methods, online RL fine-tuning requires the policy to balance exploration and exploitation, especially in sparse reward settings [20]. However, how to design a principled exploration mechanism remains elusive for conditional flows with a deterministic path.

This paper addresses these challenges head-on and proposes the first online RL approach to fine-tune a pre-trained flow matching policy. The contributions of this work are summarized as follows:

- *Algorithm Design.* We propose ReinFlow, the first online RL algorithm to stably fine-tune a family of flow matching policies, especially in very few or even in one denoising step. We train a noise injection network to convert flows to a discrete-time Markov process with Gaussian transition probabilities for exact and tractable likelihood computation. Our design allows the noise net to automatically balance exploration with exploitation; it enjoys lightweight implementation, built-in exploration, and broad applicability to various variants of flow policy, including those parameterized with Rectified Flow [34] and Shortcut Models [18].

- *Empirical Validation.* We perform extensive experiments in representative robot locomotion and manipulation tasks, with the agent receiving state or pixel observations and possibly accepting sparse rewards. Without reward shaping or scaling off-line data, our method, on average, improves the success rate over the pre-trained manipulation policy by 40.34% and increases the reward of locomotion policies by 135.36%. We achieve this improvement with a wall time reduction of 62.82% for all tasks compared to the state-of-the-art diffusion RL method DPPO [42].

- *Scientific Understanding.* We carry out systematic sensitivity analysis on design choices and key factors affecting the performance of our method, ReinFlow, including the scale of pretrained data, the number of denoising steps, the conditioning of the noise network, the level of noise and the type and intensity of different regularizations.

## 2    Related Work

In this section, we provide an outline of the relevant work. We defer a detailed introduction for several key baselines to the Appendix B.

**Online RL for improving diffusion-based policies.**    Training diffusion-based policies from demonstrations has recently achieved impressive results in a variety of robot learning tasks [3, 12, 39, 43, 50, 53, 56]. However, their performance remains highly dependent on the quality of the demonstration data. In practice, demonstrations often contain mixed or suboptimal trajectories, which limits policy effectiveness and motivates the need for online fine-tuning.

Several offline diffusion-based RL methods can be extended to online settings. For example, Diffusion Q-Learning (DQL) [54] and Implicit Diffusion Q-Learning (IDQL) [23] treat diffusion models as stochastic action policies and apply Q-learning updates. DIPO [55] adopts a similar idea, but employs a critic to directly refine the actions sampled using action gradients. Although these approaches leverage Q-function approximations to guide the diffusion actor, inaccurate Q estimates can introduce bias and destabilize training.

Alternatively, methods such as Q-Score Matching (QSM) [41] and Diffusion Policy Policy Optimization (DPPO) [42] fine-tune pre-trained diffusion policies using policy gradient techniques. Despite the growing interest in online RL for diffusion-based policies, by the time we release this work, few analogous methods exist for flow-based policies. The underlying mathematical distinctions between diffusion and flow models prevent direct adaptation of existing techniques [33], which poses additional challenges to develop effective online RL algorithms for flow policies.

**Reinforcement Learning for Flow Matching Models.** Flow matching models are more efficient than diffusion models, offering faster training, sampling, and better generalization [32, 58]. Following their success in imitation learning [8, 57] and vision tasks [32, 16, 27, 52], recent work has begun integrating them into reinforcement learning (RL).

Flow Q-Learning (FQL) [38] trains flow policies offline and distills them for fine-tuning but lacks exploration during online adaptation, leading to suboptimal convergence. ReinFlow addresses this by injecting bounded, learnable noise into flow trajectories to promote exploration.

While Flow-GRPO [33] and ORW-CFM-W2 [17] also study online RL for flow models, they focus on vision tasks rather than continuous robotic control. ReinFlow instead provides a general policy gradient framework with a compact noise network and exact likelihood computation, unlike Flow-GRPO's fixed noise or ORW-CFM-W2's reward-weighted regression approach.

## 3 Problem Formulation

**Robot Learning as a Decision Process** We formulate robot learning as an infinite-horizon Partially Observable Markov Decision Process (POMDP) in continuous state space $\mathcal{S} \in \mathbb{R}^{d_S}$, action space $\mathcal{A} \in \mathbb{R}^{d_A}$, observation space $\mathcal{O} \in \mathbb{R}^{d_O}$, with a reward discount factor $\gamma \in (0, 1)$. The robot plays in an environment with an unknown transition kernel $\mathbb{T}_{h,a}(\cdot|s)$ and an emission kernel $\mathbb{O}_h(\cdot|s)$. The interactions start with an initial state $S_0$ drawn from a distribution $\rho \in \mathscr{D}(\mathcal{S})$. In step $h \in \mathbb{Z}_{\geq 0}$, agent observes $o_h \sim \mathbb{O}_h(\cdot|s_h)$, takes an action $a_h$, before the state $s_h$ transitions to $s_{h+1} \sim \mathbb{T}_{h,a_h}(\cdot|s_h)$ with reward $r_h$. For simplicity, we study reactive policies, which map the latest observation to an action distribution. The agent aims to maximize the discounted accumulated reward $J(\pi) = \mathbb{E}^\pi \left[ \sum_{h=0}^{+\infty} \gamma^h r_h(a_h, o_h) \right]$. The Q function value function and the advantage function are defined as

$$Q_h^\pi(o_h, a_h) := \mathbb{E}^\pi \left[ \sum_{\tau=h}^{+\infty} \gamma^{\tau-h} r_\tau \mid o_h, a_h \right], \ V_h^\pi(o_h) = \mathbb{E}^\pi[Q_h^\pi(o_h, a_h)|o_h], \ A_h^\pi := Q_h^\pi - V_h^\pi \tag{1}$$

We drop the subscript $h$ for $V_h^\pi$, $Q_h^\pi$, and $A_h^\pi$ when the policy and POMDP are stationary.

**Flow Matching Models** Flow matching [32] transforms random variables from one distribution $p_0$ to another $p_1$ with flow mappings $\psi : [0, 1] \times \mathcal{X} \to \mathcal{X}$, where $X_t := \psi_t(X_0), t \in [0, 1]$. This process is associated with an ODE: $\frac{d}{dt}\psi_t(X_0) = v(t, \psi_t(X_0))$ where $X_0 \sim p_0$. Rectified flow (ReFlow) [34][2] is a simple flow model with a straight ODE path $X_t = tX_1 + (1-t)X_0$. The velocity field for Rectified Flow satisfies $v(t, X_t) = \frac{d}{dt}X_t = X_1 - X_0$, which implies that the training objective for Rectified Flow is given by

$$\hat{\theta} = \arg\min_\theta \mathbb{E}_{X_0 \sim p_0, X_1 \sim p_1, t \sim \text{Unif}[0,1]} \left[ \|X_1 - X_0 - v_\theta(t, X_t)\|_2^2 \right] \tag{2}$$

Practitioners also sample $t$ from the beta distribution [7] or the logit normal distribution [16]. [18] proposes "Shortcut Models" to further improve the generation quality of Rectified Flow at very few denoising steps by enforcing the velocity generated by two steps to align with that generated by a single step. During inference, we numerically solve the transport equation by integrating the learned velocity field: $\hat{X}_1 = X_0 + \sum_{k=0}^{K-1} v_\theta(t_i, X_{t_i})\Delta t_i$, where $K$ is the number of denoising steps, $0 = t_0 < t_1 < \ldots < t_{K-1} < 1 = t_K$ are the discretized time steps, $\Delta t_i = t_{i+1} - t_i$ is the step size.

**Flow Matching Policy** When we instantiate a flow-matching model in the action-generation setting, we obtain a flow-matching policy for robot learning. We denote by $a_h^t$ the denoised action at time $t$

---

[2]For simplicity, we only consider 1-ReFlow and use 1-ReFlow and ReFlow in this work interchangeably.

generated during the $h$-th step of the episode. $\mathcal{X}$ will be the action space $\mathcal{A}$, $p_0$ will be the standard normal distribution, and $X_t$ corresponds to the robot's denoised actions. The velocity field $v_\theta$ also conditions the observations.

Flow policies offer significantly faster inference than autoregressive and diffusion models [7, 57]. A fast policy is valuable for robot learning, as a high inference frequency enhances the robot's dexterity, and faster rollouts also reduce the wall-clock time of RL fine-tuning. Reducing denoising steps is arguably the most straightforward method to further accelerate flow policies, but when numerically solving the ODE, fewer integration steps increase the discretization error and reduce the quality of action generation [45]. This work seeks to use RL to improve flow policies with minimal denoising steps, ideally a single step, to achieve the best of both worlds: rapid inference and high success rates.

## 4 Algorithm Design

In this section, we detail the design of our algorithm "ReinFlow". We begin in Section 4.1 by introducing a methodology that precisely computes the logarithmic probability of the action in a simple closed form. This approach leverages the injection of Markov noise and eliminates the discretization error, making it applicable even with minimal denoising steps. Next, in Section 4.2, we derive a general policy gradient loss for arbitrary policies parameterized by a discrete-time Markov process, and then specialize this result in our noise-injected flow policy. Following this, Sections 4.3 and 4.4 detail the design choices for the noise injection network and the regularization techniques employed to enhance stability and promote exploration.

---

**Algorithm 1** ReinFlow

1: **Input** pre-trained flow matching policy's velocity field $v_\theta$; denoising step number $K$, discount factor $\gamma$, batch size $B$, discretization scheme $0 = t_0 < t_1 < \ldots < t_K = 1\}$ with $\Delta t_k := t_{k+1} - t_k$ regularization function $\mathcal{R}$ with intensity coefficient $\alpha \in \mathbb{R}$.
2: **Initialize** noise injection network $\theta'$.
3: **while** not converged **do**
4:     Restore last iteration's parameters: $\bar{\theta}_{\mathrm{old}} \leftarrow \mathrm{stop\_grad}([\theta, \theta'])$
5:     **Reset** environment and receive initial observation $o$.
6:     **while** not done **do**                                             ▷ Rollout policy $\pi$.
7:         Sample $a^0 \sim \mathcal{N}(0, \mathbb{I}_{d_A})$
8:         **for** denoising step k in $\{0, 1, \ldots, K-1\}$ **do**       ▷ Inject noise and integrate.
9:             $a^{k+1} \leftarrow a^k + v_\theta(t_k, a^k, o)\Delta t_k + \sigma_{\theta'}(t_k, a^k, o)\epsilon, \ \ \epsilon \sim \mathcal{N}(0, \mathbb{I}_{d_A})$
10:         **end for**
11:         Record denoised actions $a^0, a^1, \ldots, a^K$ in a buffer.
12:         Play action $a = a^K$, receive reward $r$ and done flag $d$, update observation $o$.
13:         Store $\{\mathbf{a}, o, r, d\}$ to buffer, where $\mathbf{a} := (a^0, a^1, \ldots, a^K)$
14:     **end while**
15:     Sample a batch of data $\{\mathbf{a}_i, o_i, r_i, d_i\}_{i=1}^B$ from buffer.       ▷ Optimize policy.
16:     Compute the policy's transition probability for each denoising step $k$ by Eq. (7):
17:

$$\ln \pi^{\bar{\theta}}(a_i^{k+1}|a_i^k, o_i) = \ln \mathcal{N}\left(a_i^{k+1}|a_i^k + v_\theta\left(t_k, a_i^k, o_i\right)\Delta t_k \,,\, \sigma_{\theta'}^2\left(t_k, a_i^k, o_i\right)\right)$$

18:     Compute the regularization function $\mathcal{R}$ evaluated on each tuple, denoted as $\mathcal{R}(\mathbf{a}_i, o_i; \bar{\theta}, \bar{\theta}_{\mathrm{old}})$.
19:     Call a policy gradient sub-routine, such as Alg. 2, to jointly optimize $\theta$ and $\theta'$ by Eq. (9):

$$\theta, \theta' = \underset{\theta, \theta'}{\arg\min} \frac{1}{B} \sum_{i=1}^B \left[ -A^{\bar{\theta}_{\mathrm{old}}}(o_i, a_i) \sum_{k=0}^{K-1} \ln \pi^{\bar{\theta}}(a_i^{k+1}|a_i^k, o_i) + \alpha \cdot \mathcal{R}(\mathbf{a}_i, o_i; \bar{\theta}, \bar{\theta}_{\mathrm{old}}) \right] \quad (3)$$
$$\text{where } \bar{\theta} := [\theta, \theta']$$

20: **end while**
21: **Return** fine-tuned flow matching policy's velocity field $v_\theta$.

---

**Algorithm 2** ReinFlow Subroutine for Policy Optimization (PPO implementation)

---

1: **Input** clipping range $\epsilon \in (0,1)$, policy parameters at the current iteration $\bar{\theta} := [\theta, \theta']$ and the last iteration $\bar{\theta}_{\text{old}}$, data $\{\mathbf{a}_i, o_i, r_i, d_i\}_{i=1}^{B}$, regularization function values $\{\mathcal{R}(\mathbf{a}_i, o_i; \bar{\theta}, \bar{\theta}_{\text{old}})\}_{i=1}^{B}$, with intensity $\alpha \in \mathbb{R}$.

2: Compute the advantage estimates $\widehat{A}_i := \widehat{A}(o_i, a_i^K)$ by methods such as GAE [46]

3: Jointly optimize the velocity net $\theta$ and noise net $\theta'$ by taking gradient step on the clipped surrogate loss:

$$\nabla_{\bar{\theta}} \frac{1}{B} \sum_{i=1}^{B} \left[ -\min \left( \frac{\pi_{\bar{\theta}}(\mathbf{a}_i|o_i)}{\pi_{\bar{\theta}_{\text{old}}}(\mathbf{a}_i|o_i)} \widehat{A}_i \; , \; \text{clip} \left( \frac{\pi_{\bar{\theta}}(\mathbf{a}_i|o_i)}{\pi_{\bar{\theta}_{\text{old}}}(\mathbf{a}_i|o_i)}, 1 - \epsilon, 1 + \epsilon \right) \widehat{A}_i \right) + \alpha \cdot \mathcal{R}(\mathbf{a}_i, o_i; \bar{\theta}, \bar{\theta}_{\text{old}}) \right]$$

4: **Return** updated parameters $\theta, \theta'$

---

## 4.1 Likelihood Computation over a Short Denoising Trajectory

The log probability of a policy, which quantifies action stochasticity, is essential for policy gradient methods in continuous control [22, 47]. However, computing the log probability for flow-matching policies with minimal denoising steps is challenging.

Although an exact log probability expression exists [11] for continuous-time flow models,

$$\ln p_1(\psi_1(x)) = \ln p_0(\psi_0(x)) - \int_0^1 \nabla \cdot v(t, \psi_t(x)) \, \mathrm{d}t, \quad x \sim p_0(\cdot) \tag{4}$$

in practice, we need to simulate this integral with numerical solvers and estimate the divergence [24]:

$$\widehat{\ln p_1}(x_1) = \ln p_0(x_0) - \sum_{k=0}^{K-1} \text{tr} \left[ Z^\top \partial_X v_\theta(t_i, X_{t_i}) Z \right] \Delta t_i, \tag{5}$$

where $Z$ is a zero-mean random vector with an identity covariance matrix. However, the trace estimator involves Monte-Carlo error; Simulating the integral introduces discretization error, which becomes more pronounced with larger step sizes (fewer denoising steps) [45]—a critical limitation for fast inference in robotic action generation. Treating the flow process at inference as a discrete-time Markov process can mitigate this issue. However, the intermediate variables follow a deterministic transition $p(X_{t+\Delta t} = x | X_t) = \delta(x - X_t - v_\theta(t, X_t)\Delta t)$, which renders the computation of the probability impossible.

Our approach differs from previous methods by injecting learnable noise directly into the flow model's trajectory, thereby transforming the flow into a discrete-time Markov process with closed-form transitions. During generation, actions are sampled from a normal distribution whose mean is given by the velocity field and whose standard deviation is parameterized by a noise injection network $\theta'$. For robotic policies, this yields:

$$a^0 \sim \mathcal{N}(0, \mathbb{I}_{d_A}), \;\; a^{k+1} \sim \mathcal{N}\left( \cdot \, | a^k + v_\theta(t_i, a^k, o)\Delta t_i, \;\; \sigma_{\theta'}^2(t_i, a^k, o) \right) \tag{6}$$

We condition the noise on the current denoised action and time to preserve the Markov property of the flow process. Consequently, the joint log probability of the denoising process admits the following expansion:

$$\ln \pi(a^0, \ldots, a^K | o; \theta, \theta') = \ln \mathcal{N}(0, \mathbb{I}_{d_A}) + \sum_{k=0}^{K-1} \ln \mathcal{N}\left( a^{k+1} | a_h^k + v_\theta\left( t_k, a_h^k, o \right) \Delta t_k \; , \; \sigma_{\theta'}^2\left( t_k, a^k, o \right) \right)$$

$$\tag{7}$$

where $t_k = \frac{k}{K}$ and $\Delta t_k = \frac{1}{K}$ under uniform discretization.

We treat the flow model at inference as a discrete-time process, and we have complete knowledge of the noise statistics. Unlike the empirical estimate in Eq. (5), the joint probability expression in Eq. (7) is exact, even for arbitrarily large step sizes (or equivalently, minimal denoising steps). This precision guarantees stable fine-tuning of policies, particularly for very few or even one-step inference. By avoiding trace estimation, our method eliminates errors and computational overhead associated with Monte-Carlo estimation.

## 4.2 Policy Gradient of a Discrete-time Markov Process

Although Eq. (7) provides the joint probability for the discrete-time Markov process that generates the action through denoising, it does not directly describe the marginal probability of the final action that the robot ultimately executes in the environment. Fortunately, we can establish a policy gradient theorem for policies parameterized by a discrete-time Markov process. This enables us to leverage Eq. (7) effectively to optimize the noise-injected flows.

**Theorem 4.1** (Policy Gradient Theorem for Markov Process Policy). *For a POMDP with a reactive policy $\pi^\theta$ described by a discrete-time Markov Process $o_h \to a_h^0 \to a_h^1 \to \ldots \to a_h^K = a_h$, the policy gradient is*

$$\nabla_\theta J(\pi^\theta) = \mathbb{E}^{\pi^\theta} \left[ \sum_{h=0}^{+\infty} \gamma^h A_h^{\pi^\theta}(o_h, a_h) \nabla_\theta \ln \pi^\theta(a_h^0, a_h^1, \ldots, a_h^K | o_h) \right] \tag{8}$$

*Further assuming that the policy and POMDP are stationary, the policy gradient takes the form of*

$$\nabla_\theta J(\pi^\theta) = \frac{1}{1-\gamma} \mathbb{E}_{o \sim d_\rho^{\pi^\theta}(\cdot)} \mathbb{E}_{a^0, a^1, \ldots, a^K \sim \pi^\theta(\cdot|o)} \left[ Q^{\pi^\theta}(o, a) \nabla_\theta \sum_{k=0}^{K-1} \ln \pi^\theta(a^{k+1} | a^k, o) \right] \tag{9a}$$

$$= \frac{1}{1-\gamma} \mathbb{E}_{o \sim d_\rho^{\pi^\theta}(\cdot)} \mathbb{E}_{a^0, a^1, \ldots, a^K \sim \pi^\theta(\cdot|o)} \left[ A^{\pi^\theta}(o, a) \nabla_\theta \sum_{k=0}^{K-1} \ln \pi^\theta(a^{k+1} | a^k, o) \right] \tag{9b}$$

*where $d_\rho^\pi$ is the discounted observation visitation frequency, defined as*

$$d_\rho^\pi(o) := (1-\gamma) \mathbb{E}_{s_1 \sim \rho(\cdot)} \left[ \sum_{h=0}^{+\infty} \gamma^h \, p(o_h = o | s_1; \pi) \right] \tag{10}$$

Theorem 4.1 builds on Theorem B.1 in [40] and established results in policy gradient theory [2]. We provide the proofs in Appendix A.

Using Eq. (7), we enable the application of various modern deep policy gradient algorithms to optimize a policy parameterized by a discrete-time Markov process, including the noise-injected flows introduced in Section 4.1, which are of particular interest. Specifically, we can optimize the policy using either Eq. (9a) or Eq. (9b). In this work, we implement Eq. (9b) with the clipped surrogate loss [47] due to its stability.

By combining Eq. (7) with Eq. (9), we derive our algorithm, "ReinFlow", outlined in Alg. 1. We also present one possible policy optimization subroutine in Alg. 2. In Alg. 1, we denote the denoised action sequence $a^0, \ldots, a^K$ as a boldfaced $\mathbf{a}$ and represent the combined parameters of the velocity and noise networks as $\bar{\theta} = [\theta, \theta']$.

We also note that, beyond the implementations in Alg. 1 and Alg. 2, we can optimize Eq. (9b) by replacing Eq. (3) with:

$$\theta, \theta' = \operatorname*{argmin}_{\theta, \theta'} \frac{1}{B} \sum_{i=1}^{B} \left[ -Q^{\bar{\theta}_{\text{old}}}(o_i, a_i) \sum_{k=0}^{K-1} \ln \pi^{\bar{\theta}}(a_i^{k+1} | a_i^k, o_i) + \alpha \cdot \mathcal{R}(\mathbf{a}_i, o_i; \bar{\theta}) \right], \text{where } \bar{\theta} := [\theta, \theta'] \tag{11}$$

and employ off-policy methods, such as SAC [22], to update the policy.

In practice, we adopt "action chunking" during policy execution [29], in which the policy outputs multiple actions in a batch given an observation. We explain how action chunking affects likelihood computation in Appendix A.2.

## 4.3 Noise Injection Network

The noise has the same shape of the joint torques and is applied to the whole denoising process of action generation. With the noise, we characterize the policy's probability in closed form. The price we take is a slight increase in noise net parameters, which is only a fraction of the pre-trained policy, as indicated by Table 2.

During fine-tuning, the noise injection network $\theta'$ is trained with the velocity network $\theta$ to add diversity to the sample path for better exploration. The loss function for $\theta'$ is the loss of the policy gradient (Eq. (9)), the same as for the net velocity. After fine-tuning, we discard the noise net $\theta'$ and recover the flow matching policy, which is still made up of deterministic maps. By optimizing the noise net, the agent automatically adjusts her exploration level over the course of training.

The noise network could condition on $o$, $(o, t)$, or even output fixed constants. We provide a comparative study in Section 6 to analyze how the noise conditions affect ReinFlow's performance. When the noise injection network is learnable, it uses features from the pre-trained flow policy to save parameters and ensure consistency.

We determine the noise limit using a set of key hyperparameters that vary according to the mechanical structural constraints of each joint. We study the effect of the noise bounds in Section 5. Empirically, we find that when the success rate converges to 100%, the noise level naturally decays. We can also manually control the noise level by adjusting the limits, or even increase the standard deviation of the noise with regularization of the entropy to promote exploration (see Section 4.4).

### 4.4 Regularization

ReinFlow also supports various ways to regularize the fine-tuned policy.

**Wasserstein-2 ($W_2$) Regularization.** One approach is to adopt Wasserstein regularization [28, 38], which constrains the Wasserstein-2 distance from the fine-tuned policy to the pre-trained policy. Studies show that $W_2$ regularization could improve training stability for generative models [4, 28]. In practice, we minimize a tractable upper bound on this distance to enforce such constraint:

$$
\begin{aligned}
\mathcal{R}_{\mathrm{W}_2}(\theta, \theta_{\mathrm{old}}) &= \mathbb{E}_o \mathbb{E}_{a \sim \pi_\theta(\cdot|o), a_{\mathrm{old}} \sim \pi_{\theta_{\mathrm{old}}}(\cdot|o)} \left[ \tfrac{1}{2} \| a - a_{\mathrm{old}} \|_2^2 \right] \\
&\geq \mathbb{E}_o \left[ \inf_{\lambda \in \Lambda} \mathbb{E}_{x, y \sim \lambda_o} \left[ \tfrac{1}{2} \| x - y \|_2^2 \right] \right] := \mathbb{E}_o \left[ W_2^2(\pi_{\theta_{\mathrm{old}}}(\cdot|o), \pi_\theta(\cdot|o)) \right]
\end{aligned}
\tag{12}
$$

In Eq. (12), $\Lambda_o$ indicates the set of distributions over $\mathcal{A}^2$ whose marginals are $\pi_{\theta_{\mathrm{old}}}(\cdot|o)$ and $\pi_\theta(\cdot|o)$. To implement Eq. (12), we replace the expectations with the sample mean over a batch of data. Inspired by [38], we integrate $a$ and $a_{\mathrm{old}}$ from the same starting noise $a_0 \sim \mathcal{N}(0, \mathbb{I}_{d_A})$, to control the stochasticity of the initial denoise action. We also note that when computing $\mathcal{R}_{W_2}$, we do not inject noise when sampling $a$ from $\pi_\theta$.

**Entropy Regularization.** Another technique is entropy regularization, which theory has shown accelerates convergence [1, 10] and encourages exploration [22] for simple policy classes. For a flow matching policy parameterized with a discrete-time Markov process, we adopt the negative per-symbol entropy rate (or block entropy) [48] as the entropy regularizer. For a stationary POMDP and policy, we define the regularizer as

$$
\begin{aligned}
\mathcal{R}_{\mathbf{h}}(\bar{\theta}) &:= -\frac{1}{K+1} \mathbb{E} \left[ \mathbf{h}(a^0, a^1, \ldots, a^K | o, \bar{\theta}) \right] \\
&= -\frac{1}{K+1} \mathbb{E} \left[ \mathbf{h} \left( \mathcal{N}(0, \mathbb{I}_{d_A}) \right) + \sum_{k=0}^{K-1} \mathbf{h} \left( \mathcal{N} \left( a^k + v_\theta(t_k, a^k, o) \Delta t_k, \sigma_{\theta'}^2(t_k, a^k, o) \right) \right) \right]
\end{aligned}
$$

Here, the last step is due to the definition of the noise-injected flow process in Eq. (6). $\mathbf{h}$ is the differential entropy operator [14], of which normal distributions possess a simple closed-form expression specified in Section 3. According to [48], the per-symbol entropy measures the Shannon entropy of a finite symbol sequence with long-range correlations. By minimizing $\mathcal{R}_{\mathbf{h}}$, we promote the agent to seek more diverse actions, enhancing exploration.

We carry out experiments in Section 5 to compare the effects of different regularizations on the performance of ReinFlow. By default, we adopt entropy regularization in Algorithm 1 for state-input tasks and do not adopt regularization for visual manipulation tasks.

## 5 Experiments

We run simulated robot learning experiments to test the effectiveness and flexibility of our method, ReinFlow. We aim to adopt ReinFlow to significantly enhance the success rate of pre-trained flow

matching policies trained on mediocre expert data. We also managed to fine-tune at very few or even one denoising step for Rectified Flow (indicated by "ReinFlow-R") and Shortcut Models (indicated by "ReinFlow-S"), where fine-tuning the two types of pre-trained models shares the same training hyperparameters.

We adopt a PPO-based implementation of our algorithm due to its stability. Although alternative implementations may offer higher sample efficiency, we prioritize wall-time efficiency in simulated environments over sample cost. Exploring more sample-efficient RL algorithms for ReinFlow, especially in real-world scenarios where data collection is expensive, is an interesting direction for future work.

We compare ReinFlow against various RL methods for fine-tuning diffusion and flow-based policies, particularly DPPO and FQL. DPPO is a strong online RL algorithm for diffusion policies, developed under a bilevel MDP formulation [42]. FQL represents state-of-the-art offline RL for flow-matching policies and applies to offline-to-online fine-tuning. Comparisons with other baselines are provided in the Appendix E.

## 5.1 Environment Setup and Data Curation

We compare the algorithms in locomotion and manipulation tasks. In locomotion, the agent receives state input and a dense reward, similar to sim2real RL for legged robots [44]. We also consider manipulation tasks, where agents receive pixel and/or state inputs with sparse rewards.

**OpenAI Gym [9].** We fine-tuned flow matching policies with ReinFlow in "Hopper", "Walker2d", "Ant" and "Humanoid" where these tasks are listed in ascending difficulty, and the last two involve high-dimensional state inputs and are considered very challenging continuous control problems. Expert data are medium- or medium-expert-level demonstrations collected from the D4RL dataset [19]. [3]

**Franka Kitchen [21].** In Franka Kitchen, a Franka robot learns long-horizon multitask planning by completing four state-based manipulation tasks sequentially. We pre-train the models with human teleoperated data with complete, mixed, or partial demonstrations of the four tasks.

**Robomimic [36].** We adopt Robomimic visual manipulation tasks: PickPlaceCan (Can), NutAssemblySquare (Square), and TwoArmTransport (Transport). Data are collected via human teleportation and processed following DPPO [42], containing fewer and lower-quality data than proficient human.

## 5.2 Experiments

ReinFlow demonstrates strong training stability with a significant increase in the success rate or success rate in all tasks. In Gym and Franka Kitchen benchmarks, it achieves the best overall efficiency and performance improvement.

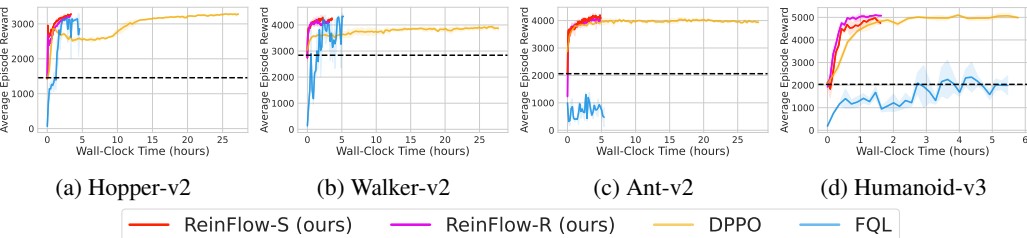

Figure 1: Wall time efficiency in OpenAI Gym. Dashed lines indicate the behavior cloning level.

Across three Robomimic visual manipulation tasks, ReinFlow-S and ReinFlow-R improve the success rate of the pre-trained policy by an average of 45.77% . ReinFlow achieves success rates comparable to DPPO, requiring significantly fewer fine-tuning steps and less wall-clock time. Notably, it uses fewer denoising steps: just *one* in can and square, and a four-step flow in transport, compared to the five-step DDIM used by DPPO.

---

[3]Except for "Humanoid" task, where the data is sampled from our own pre-trained SAC agent.

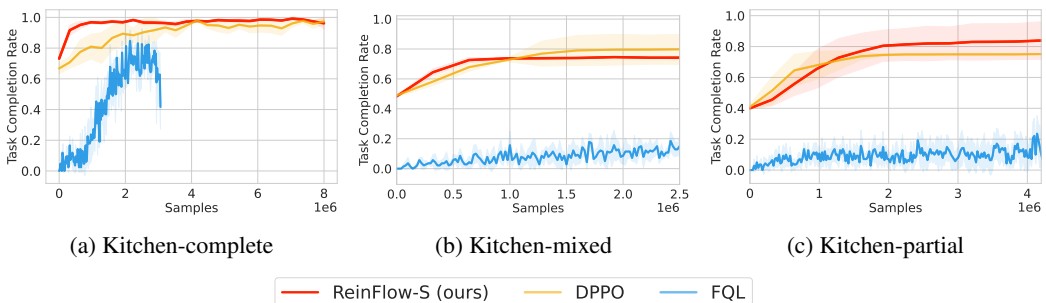

| (a) Kitchen-complete | (b) Kitchen-mixed | (c) Kitchen-partial |

Figure 2: Task completion rates of state-input manipulation tasks in Franka Kitchen

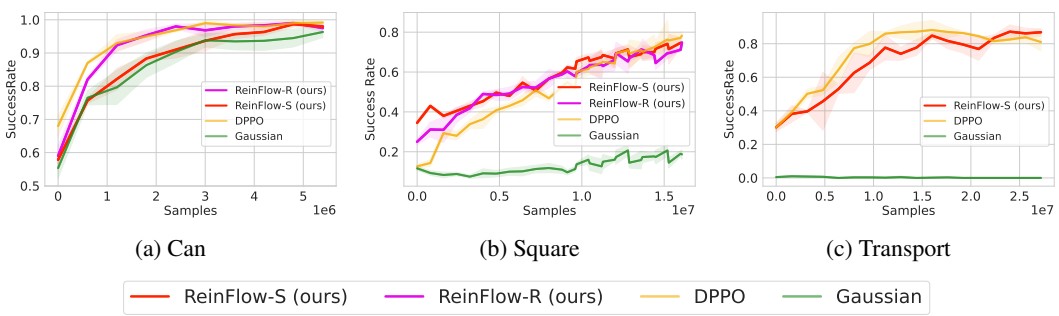

| (a) Can | (b) Square | (c) Transport |

Figure 3: Success rates in visual manipulation tasks in Robomimic.

# 6 The Design Choice and Key Factors Affecting ReinFlow

This section analyzes how the pre-trained model and denoising steps affect our algorithm. We also study the effects of the noise level, the type, and the intensity of regularization.

**Scaling.** We fine-tune flow matching policies trained on datasets with different numbers of episodes and test the performance of pre-trained and fine-tuned models at different denoising steps. Fig 4a reveals that scaling inference steps and/or pre-training data quantity does not consistently improve the reward, which is consistent with the findings in [31]. However, ReinFlow consistently improves the success rate or reward over the pre-trained policies regardless of the pre-training scale.

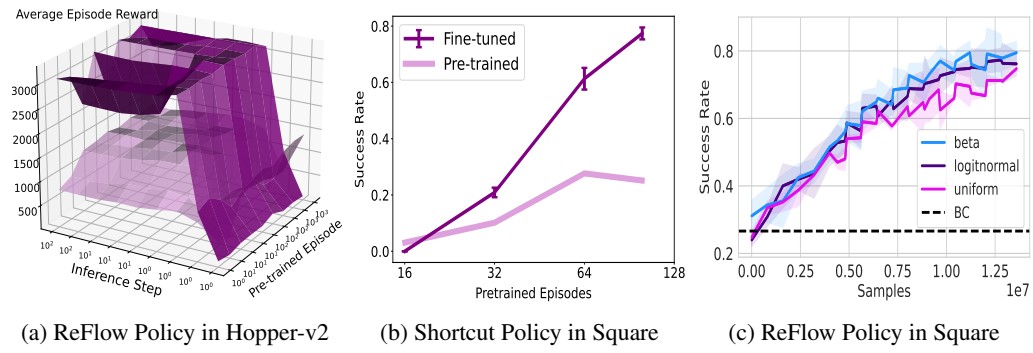

| (a) ReFlow Policy in Hopper-v2 | (b) Shortcut Policy in Square | (c) ReFlow Policy in Square |

Figure 4: RL offers an orthogonal scaling path beyond data or inference. The gain is invariant to denoising steps—at 4 steps in Hopper and 1 in Square.

**Flow Matching's Time Distribution.** The effectiveness of ReinFlow is not affected by altering the time sampling distribution of the pre-trained flow matching policy. However, the beta distribution is slightly stronger when fine-tuning in one denoising step, as in the case in Fig. 4c.

**Noise Network Inputs.** The inputs of the noise injection network affect the performance of fine-tuned flow policies. As shown in Fig. 5, conditioning both on observations and time often yields a higher success rate, as this approach allows the noise network to learn how to create more diverse actions by altering the noise intensity at different denoising steps.

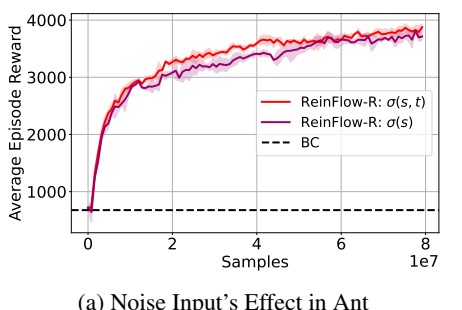

(a) Noise Input's Effect in Ant

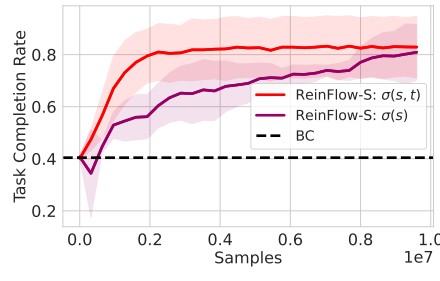

(b) Noise Condition's Effect in Kitchen-partial

Figure 5: Conditioning on state and time yields a higher success rate than only conditioning on states.

**Noise Level and Exploration.** The noise magnitude is the key factor that influences the performance of ReinFlow. Fig. 6a shows small noise leads to limited exploration, while moderate noise enables rapid improvement, up to three times higher rewards. Beyond this threshold, performance becomes less sensitive to noise. Reducing noise proves beneficial for visuomotor policies, precision-critical tasks, longer denoising chains, and weakly pret-trained models.

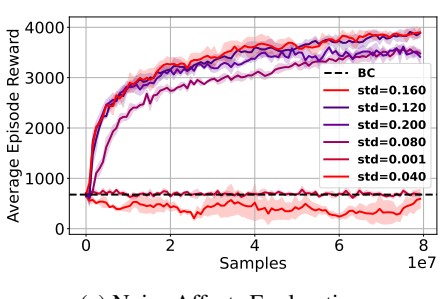

(a) Noise Affects Exploration

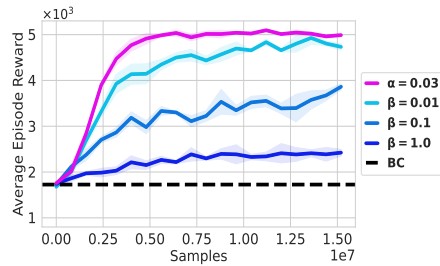

(b) $W_2$ (blue) and Entropy (red) Regularization

Figure 6: Effect of noise and regularization in Ant-v0 (left) and Humanoid-v3 (right)

**Regularization and Exploration.** ReinFlow supports various regularizations. Adding entropy regularization, especially in locomotion tasks, is generally more effective than constraining the policy with a $W_2$ regularizer. As shown in Fig. 6b, reducing the $W_2$ coefficient $\beta$ allows the policy to surpass the behavior cloning baseline and learn more robust actions, approaching the performance achieved with entropy regularization ($\alpha = 0.03$). This also helps explain why the offline RL method FQL, which enforces $W_2$ constraints during training, underperforms compared to our online approach.

# 7 Conclusion, Limitations, and Future Work

This work introduces ReinFlow, the first online reinforcement learning (RL) framework that stably fine-tunes a family of flow matching policies for continuous robotic control. In state-input locomotion and visual manipulation tasks, ReinFlow surpasses existing methods that fine-tune diffusion or flow models using RL, while reducing wall-clock time by over 50% compared to the state-of-the-art diffusion RL algorithm. We conducted a sensitivity analysis to identify the key factors affecting ReinFlow's performance.

ReinFlow's current implementation has several limitations. Although the on-policy design saves wall-time with parallelism, future work should explore a sample-efficient implementation and adapting ReinFlow to real-world RL. Another open direction is to reduce its sensitivity to noise magnitude, auto-tune or remove these hyperparameters. Lastly, our current experiments use relatively small networks, and scaling ReinFlow to large flow-based vision-language-action (VLA) models remains an exciting challenge. We leave these to future work.

**Acknowledgements** This work was supported by National Natural Science Foundation of China (No.62406159, 62325405), Postdoctoral Fellowship Program of CPSF under Grant Number (GZC20240830, 2024M761676), China Postdoctoral Science Special Foundation 2024T170496, and Beijing Zhongguancun Academy Project C20250301. The authors are grateful to Shu'ang Yu for reviewing the earlier versions of the paper. We also thank Feng Gao, Cheng Yin, and Ningyuan Yang for many fruitful discussions and comments.

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

Figure 7: Fine-tuning a flow matching policy with online RL algorithm ReinFlow (Alg. 1).

Through interactions with the environment, the robot collects visual and proprioceptive signals, from which a pre-trained policy extracts features and outputs the velocity field of the following action, $v_\theta$. A noise injection network $\sigma_{\theta'}$ shares the extracted features with $v_\theta$ and outputs a Gaussian noise that smoothens the flow's deterministic ODE path, converting flows to a discrete-time Markov process with Gaussian transition probabilities. The noise injection yields an exact and straightforward likelihood expression at any denoising steps, which is friendly for policy gradient optimization. The noise injection net $\sigma_{\theta'}$, surrounded by the dot lines, co-trained with $v_\theta$ but will be discarded after fine-tuning. The size of $\sigma_{\theta'}$ is only a fraction of the pre-trained flow policy. We outline our findings on a webpage: https://reinflow.github.io/. In what follows, we provide the theoretical background of our algorithm, report the findings omitted by the main text, and elaborate on the implementation details required to reproduce our experimental results.

# A  Theoretical Support

## A.1  Proof of Theorem 4.1

In this section, we prove Theorem 4.1, the policy gradient theorem for discrete-time Markov process policies.

For notation simplicity, we only consider an infinite-horizon POMDP with a reactive policy. We obtain the result for the finite-horizon setting by imposing $r_h = 0$ for $h$ larger than the finite horizon $H$. For the reader's convenience, we recall several fundamental definitions of RL. The objective function for RL is given by $J(\pi) = \mathbb{E}^\pi \left[ \sum_{h=0}^{+\infty} \gamma^h r_h(o_h, a_h) \right]$, The value function, Q function, and the advantage functions are defined as

$$
\begin{aligned}
V_h^\pi(o_h) :=& \mathbb{E}^\pi \left[ \sum_{\tau=h}^{+\infty} \gamma^{\tau-h} r_\tau(a_\tau, o_\tau) \mid o_h \right] \\
Q_h^\pi(o_h, a_h) :=& \mathbb{E}^\pi \left[ \sum_{\tau=h}^{+\infty} \gamma^{\tau-h} r_\tau(a_\tau, o_\tau) \mid o_h, a_h \right] \quad A_h^\pi(o_h, a_h) := Q_h^\pi(o_h, a_h) - V^\pi(o_h)
\end{aligned}
\tag{13}
$$

We first show that we can express the policy gradient for POMDP in terms of the advantage function and the action log probability's gradient:

$$
\nabla_\theta J(\pi^\theta) = \mathbb{E}^{\pi^\theta} \left[ \sum_{\tau=0}^{+\infty} \gamma^\tau A_\tau^{\pi^\theta}(o_\tau, a_\tau) \nabla_\theta \ln \pi_\theta(a_\tau | o_\tau) \right]
\tag{14}
$$

*Remark* A.1. In Eq. (14), we use $\theta$ to indicate general policy parameters. $\theta$ should be understood as $\bar{\theta} = [\theta, \theta']$, that is, the combination of the velocity and noise nets when we instantiate the policy as a noise-injected flow matching policy.

*Proof.*

$$\nabla_\theta J(\pi^\theta) = \sum_{h=0}^{+\infty} \gamma^h \int_{\mathcal{O} \times \mathcal{A}^h} r_h(o_h, a_h) \cdot \nabla_\theta p(o_{1:h}, a_{1:h}|\pi^\theta)$$

$$= \sum_{h=0}^{+\infty} \gamma^h \int_{\mathcal{O} \times \mathcal{A}^h} r_h(o_h, a_h) \cdot \nabla_\theta \exp \ln p(o_{1:h}, a_{1:h}|\pi^\theta)$$

$$= \sum_{h=0}^{+\infty} \gamma^h \int_{\mathcal{O} \times \mathcal{A}^h} r_h(o_h, a_h) \cdot p(o_{1:h}, a_{1:h}|\pi^\theta) \nabla_\theta \ln p(o_{1:h}, a_{1:h}|\pi^\theta)$$

$$\overset{(i)}{=} \sum_{h=0}^{+\infty} \gamma^h \int_{\mathcal{O} \times \mathcal{A}^h} r_h(o_h, a_h) \cdot p(o_{1:h}, a_{1:h}|\pi^\theta) \sum_{\tau=1}^{h} \nabla_\theta \ln \pi_\theta(a_\tau|o_\tau)$$

$$= \mathbb{E}^{\pi^\theta} \left[ \sum_{h=0}^{+\infty} \gamma^h r_h(o_h, a_h) \sum_{\tau=1}^{h} \nabla_\theta \ln \pi_\theta(a_\tau|o_\tau) \right]$$

$$= \mathbb{E}^{\pi^\theta} \left[ \sum_{\tau=0}^{+\infty} \sum_{h=\tau}^{+\infty} \gamma^h r_h(o_h, a_h) \nabla_\theta \ln \pi_\theta(a_\tau|o_\tau) \right] \quad \text{// Change summation order}$$

$$= \mathbb{E}^{\pi^\theta} \left[ \sum_{\tau=0}^{+\infty} \gamma^\tau \nabla_\theta \ln \pi_\theta(a_\tau|o_\tau) \sum_{h=\tau}^{+\infty} \gamma^{h-\tau} r_h(o_h, a_h) \right]$$

$$= \mathbb{E}^{\pi^\theta} \left[ \sum_{\tau=0}^{+\infty} \gamma^\tau \nabla_\theta \ln \pi_\theta(a_\tau|o_\tau) \mathbb{E} \left[ \sum_{h=\tau}^{+\infty} \gamma^{h-\tau} r_h(o_h, a_h) \mid a_\tau, o_\tau \right] \right]$$

$$= \mathbb{E}^{\pi^\theta} \left[ \sum_{\tau=0}^{+\infty} \gamma^\tau Q_\tau^{\pi^\theta}(o_\tau, a_\tau) \nabla_\theta \ln \pi_\theta(a_\tau|o_\tau) \right] \quad \text{// Definition in Eq. (13)}$$

$$= \mathbb{E}^{\pi^\theta} \left[ \sum_{\tau=0}^{+\infty} \gamma^\tau Q_\tau^{\pi^\theta}(o_\tau, a_\tau) \nabla_\theta \ln \pi_\theta(a_\tau|o_\tau) \right] - \underbrace{\sum_{\tau=0}^{+\infty} \frac{\gamma^\tau V^{\pi^\theta}}{\pi_\theta(a_\tau|o_\tau)} \nabla_\theta \int_{\mathcal{A}} da_\tau \pi_\theta(a_\tau|o_\tau)}_{=0}$$

$$= \mathbb{E}^{\pi^\theta} \left[ \sum_{\tau=0}^{+\infty} \gamma^\tau A_\tau^{\pi^\theta}(o_\tau, a_\tau) \nabla_\theta \ln \pi_\theta(a_\tau|o_\tau) \right] \quad \text{// Definition in Eq. (13)}$$

where (i) is due to the Markov property:

$$\begin{aligned}
&\nabla_\theta \ln p(o_{1:t}, a_{1:t}|\pi^\theta) \\
=&\nabla_\theta \left( \ln \left( \rho(s_1) \cdot \mathbb{O}_1(o_1|s_1) \cdot \pi_\theta(a_1|o_1) \cdot \mathbb{T}_h(s_2|s_1, a_1) \cdot \ldots \cdot \mathbb{O}_h(o_h|a_h) \cdot \pi_\theta(a_h|o_h) \right) \right) \\
=&\nabla_\theta \ln \pi_\theta(a_1|o_1) + \nabla_\theta \ln \pi_\theta(a_2|o_2) + \ldots + \nabla_\theta \ln \pi_\theta(a_t|o_t)
\end{aligned} \quad (15)$$

$\square$

Next, we extend Theorem 1 in [40] and study the case when the action is generated via a Markov Process. The action probability is expressed by

$$\begin{aligned}
\pi_\theta(a_h|o_h) &= \int_{\mathcal{A}^K} da_h^0 da_h^1 \ldots da_h^{K-1} \, \pi_\theta(a_h^0, a_h^1, \ldots, a_h^K|o_h) \\
&= \int_{\mathcal{A}^K} da_h^0 da_h^1 \ldots da_h^{K-1} \, \pi_\theta(a_h^0|o_h) \cdot \prod_{k=0}^{K-1} \pi_\theta(a_h^{k+1}|a_h^k, o_h)
\end{aligned} \quad (16)$$

where we write $a_h = a_h^K$. Bringing Eq. (16) to Eq. (14), we obtain

$$
\begin{aligned}
&\nabla_\theta J(\pi^\theta)\\
=&\mathbb{E}^{\pi^\theta}\left[\sum_{\tau=0}^{+\infty}\gamma^\tau A_\tau^{\pi^\theta}(o_\tau,a_\tau)\nabla_\theta\ln\pi_\theta(a_\tau|o_\tau)\right]\\
=&\sum_{\tau=0}^{+\infty}\gamma^\tau\int_\mathcal{A}\mathrm{d}a_\tau\cancel{\pi_\theta(a_\tau|o_\tau)}A_\tau^{\pi^\theta}(o_\tau,a_\tau)\frac{1}{\cancel{\pi_\theta(a_\tau|o_\tau)}}\nabla_\theta\pi_\theta(a_\tau|o_\tau)\\
=&\sum_{\tau=0}^{+\infty}\gamma^\tau\int_\mathcal{A}\mathrm{d}a_\tau^K A_\tau^{\pi^\theta}(o_\tau,a_\tau)\cdot\left[\int_{\mathcal{A}^K}\mathrm{d}a_\tau^0\mathrm{d}a_\tau^1\ldots\mathrm{d}a_\tau^{K-1}\nabla_\theta\left(\pi_\theta(a_\tau^0|o_\tau)\cdot\prod_{t=0}^{K-1}\pi_\theta(a_\tau^{t+1}|a_\tau^t,o_\tau)\right)\right]\\
=&\sum_{\tau=0}^{+\infty}\gamma^\tau\int_{\mathcal{A}^{K+1}}\mathrm{d}a_\tau^0\cdots\mathrm{d}a_\tau^K\cdot A_\tau^{\pi^\theta}(o_\tau,a_\tau)\cdot\left[\nabla_\theta\exp\ln\pi_\theta(a_\tau^0,a_\tau^1,\ldots,a_\tau^K|o_\tau)\right]\\
=&\sum_{\tau=0}^{+\infty}\gamma^\tau\int_{\mathcal{A}^{K+1}}\mathrm{d}a_\tau^0\cdots\mathrm{d}a_\tau^K\cdot A_\tau^{\pi^\theta}(o_\tau,a_\tau)\cdot\left[\pi_\theta(a_\tau^0,a_\tau^1,\ldots,a_\tau^K|o_\tau)\cdot\nabla_\theta\ln\pi_\theta(a_\tau^0,a_\tau^1,\ldots,a_\tau^K|o_\tau)\right]\\
=&\mathbb{E}^{\pi^\theta}\left[\sum_{\tau=0}^{+\infty}\gamma^\tau A_\tau^{\pi^\theta}(o_\tau,a_\tau)\nabla_\theta\ln\pi_\theta(a_\tau^0,a_\tau^1,\ldots,a_\tau^K|o_\tau)\right]\\
=&\mathbb{E}^{\pi^\theta}\left[\sum_{\tau=0}^{+\infty}\gamma^\tau A_\tau^{\pi^\theta}(o_\tau,a_\tau)\nabla_\theta\sum_{k=0}^{K-1}\ln\pi_\theta(a_\tau^{k+1}|a_\tau^k,o_\tau)\right]
\end{aligned}
\tag{17}
$$

In what follows, we consider the case where the POMDP has stationary transition kernels, and the policy is reactivate and stationary. This setting makes $A^\pi, Q^\pi, V^\pi$ time independent, so we drop their subscripts $h$ for brevity:

$$
\nabla_\theta J(\pi^\theta) = \mathbb{E}_{o,a^0,a^1,\ldots,a^K}^{\pi^\theta}\left[\sum_{\tau=0}^{+\infty}\gamma^\tau A^{\pi^\theta}(o,a^K)\nabla_\theta\sum_{k=0}^{K-1}\ln\pi_\theta(a^{k+1}|a^k,o)\right]
\tag{18}
$$

To proceed from the RHS of Eq. (18), we first show that for any function $f(\cdot,\cdot):\mathcal{O}\times\mathcal{A}\to\mathbb{R}$ and a reactive, stationary policy $\pi$, the following relation holds:

$$
\mathbb{E}^\pi\left[\sum_{h=0}^{+\infty}\gamma^h f(o_h,a_h)\right] = \frac{1}{1-\gamma}\mathbb{E}_{o\sim d_\rho^\pi(\cdot)}\left[\mathbb{E}_{a\sim\pi(\cdot|o)}\left[f(o,a)\right]\right]
\tag{19}
$$

where we take the expectation of observation $o$ with respect $d_\rho^\pi(o)$, the discounted average visitation frequency to that observation given initial state distribution. Concretely speaking, $d_\rho^\pi(o)$, which we call the "observation visitation measure", is defined by

$$
d_{s_1}^\pi(o) := (1-\gamma)\sum_{h=0}^{+\infty}\gamma^h\int_\mathcal{O}\mathrm{d}o\, p(o_h=o|s_1;\pi)
\tag{20a}
$$

$$
d_\rho^\pi(o) := \mathbb{E}_{s_1\sim\rho(\cdot)}\left[d_{s_1}^\pi(o)\right]
\tag{20b}
$$

The MDP version for this result can be found in classical RL theory textbooks, such as Eq. (0.10) on page 27 of [1]. Below, we extend the MDP result to POMDPs.

*Proof.*

$$\text{LHS} = \int_{\mathcal{S}} ds_1 \, \rho(s_1) \sum_{h=0}^{+\infty} \gamma^h \int_{\mathcal{O} \times \mathcal{A}} do_h da_h \; p(o_h, a_h | s_1) f(o_h, a_h)$$

$$= \int_{\mathcal{S}} ds_1 \, \rho(s_1) \sum_{h=0}^{+\infty} \gamma^h \int_{\mathcal{O} \times \mathcal{A}} do_h da_h \; p(o_h | s_1) \pi_h(a_h | o_h) f(o_h, a_h)$$

$$= \frac{1}{1-\gamma} \int_{\mathcal{S}} ds_1 \, \rho(s_1) \, (1-\gamma) \sum_{h=0}^{+\infty} \gamma^h \int_{\mathcal{O}} do_h \, p(o_h | s_1) \int_{\mathcal{A}} da_h \, \pi_h(a_h | o_h) f(o_h, a_h)$$

$$= \frac{1}{1-\gamma} \mathbb{E}_{s_1 \sim \rho(\cdot)} (1-\gamma) \sum_{h=0}^{+\infty} \gamma^h \int_{\mathcal{O}} do \, p(o_h = o | s_1) \mathbb{E}_{a \sim \pi_h(\cdot | o_h)} f(o, a) \quad \text{//Re-labeling}$$

$$:= \frac{1}{1-\gamma} \mathbb{E}_{s_1 \sim \rho(\cdot)} \mathbb{E}_{o \sim d_{s_1}(\cdot)} \mathbb{E}_{a \sim \pi_h(\cdot | o_h)} f(o, a) \quad \text{//Eq. (20a)}$$

$$:= \frac{1}{1-\gamma} \mathbb{E}_{o \sim d_{\rho}(\cdot)} \mathbb{E}_{a \sim \pi_h(\cdot | o_h)} f(o, a) \quad \text{//Eq. (20b)}$$

$$= \frac{1}{1-\gamma} \mathbb{E}_{o \sim d_{\rho}(\cdot)} \mathbb{E}_{a \sim \pi(\cdot | o)} f(o, a) \quad \text{//Stationary Policy}$$

$$\square$$

Instantiate the function $f$ in Eq. (19) as the product of the advantage function times and the gradient of the joint log probability, we arrive at the following result by plugging Eq. (19) into Eq. (18):

$$\nabla_{\theta} J(\pi^{\theta}) = \frac{1}{1-\gamma} \mathbb{E}_{o \sim d_{\rho}^{\pi^{\theta}}(\cdot)} \mathbb{E}_{a^0, a^1, \ldots, a^K \sim \pi^{\theta}(\cdot | o)} \left[ A^{\pi^{\theta}}(o, a) \nabla_{\theta} \sum_{k=0}^{K-1} \ln \pi_{\theta}(a^{k+1} | a^k, o) \right] \quad (21)$$

where $a$ stands for $aK$. We take the expectation of observation concerning its visitation measure and the expectation of intermediate actions for the distribution induced by the policy's Markov Process.

We remind the reader that Eq. (21) holds only for POMDPs with a stationary and reactive policy.

## A.2 Action Chunking

In practice, we implement a flow matching policy with action chunking, which slightly alters how we compute the policy's log probability. Our robots interact with the environment in a manner similar to Diffusion Policy [12], where the agent receives one observation, outputs a sequence of actions, executes each action, collects rewards per step, and accumulates them for optimization.

This interaction protocol implies that actions in a chunk are fixed after the first observation and remain unaffected by later observations that may change during execution. Thus, actions within a chunk are conditionally independent, given the initial observation. In our setup, we flatten the actions of the chunk into a single vector as the actor's output, ensuring that the actions in the chunk are independent given the network's inputs, which include the action chunk at the last denoising step, the observation, and the denoising time. Hence, the log probability of a chunk of size $C$ is the sum of the log probabilities of its internal actions. Formally:

$$\begin{aligned}
&\ln \pi_{\bar{\theta}} \left( a_{h:h+C-1}^{k+1} | t_k, a_{h:h+C-1}^k, o_h \right) \\
&= \sum_{c=0}^{C-1} \ln \pi_{\bar{\theta}} \left( a_{h+c}^{k+1} | t_k, a_{h:h+C-1}^k, o_h \right) \quad \text{//Conditional Independence} \\
&= \sum_{c=0}^{C-1} \ln \mathcal{N} \left( a_{h+c}^{k+1} \middle| a_{h+c}^k + [v_{\theta}]_{h+c} \cdot \Delta t_k, \, [\sigma_{\theta'}]_{h+c}^2 \right)
\end{aligned} \quad (22)$$

In Eq. (22), the terms $v_{\theta}$ and $\sigma_{\theta'}$ are conditioned on $(t_k, a_{h:h+C-1}^k, o_h)$. We use $[u]_i$ to denote the $i$-th element of vector $u$, and $u_{i:j}$ to represent the sub-vector formed by concatenating the $i$-th to the $j$-th coordinates of $u$.

# B  Extended Related Work

**Diffusion Policy Policy Optimization [42]**  Diffusion Policy Policy Optimization (DPPO) trains DDIM policies with PPO. They design the algorithm on top of a bi-level MDP formulation, which involves calculating the advantage function for the denoised actions.

$$\mathcal{L}_\theta = \mathbb{E} \min \left( \hat{A}^{\bar{\pi}_{\theta_{\text{old}}}} \left( \bar{s}_{\bar{t}}, \bar{a}_{\bar{t}} \right) \frac{\bar{\pi}_\theta \left( \bar{s}_{\bar{t}}, \bar{a}_{\bar{t}} \right)}{\bar{\pi}_{\theta_{\text{old}}} \left( \bar{s}_{\bar{t}}, \bar{a}_{\bar{t}} \right)}, \hat{A}^{\pi_{\theta_{\text{old}}}} \left( \bar{s}_{\bar{t}}, \bar{a}_{\bar{t}} \right) \text{clip} \left( \frac{\bar{\pi}_\theta \left( \bar{s}_{\bar{t}}, \bar{a}_{\bar{t}} \right)}{\bar{\pi}_{\theta_{\text{old}}} \left( \bar{s}_{\bar{t}}, \bar{a}_{\bar{t}} \right)}, 1 - \varepsilon, 1 + \varepsilon \right) \right) \tag{23}$$

The term $\bar{t} = \bar{t}(t, k)$ represents the $k$-th denoising step in the $t$-th interaction. They define the advantage function at these denoising steps by applying an additional discount factor, $\gamma_{\text{denoise}}^k$, to the advantage function of the action taken.

DPPO does not provide a theoretical guarantee for its design in Eq. (23). Computing the advantage function at denoised steps also slows down computation. Our method does not perform such calculations, and we derive our algorithm from rigorous reinforcement learning theory for POMDPs elaborated in Appendix A.

Empirically, Section 5 shows our method, ReinFlow, outperforms DPPO in almost all tasks in Gym and Franka Kitchen while reducing the denoising step count from 10 in DPPO to 4 in ReinFlow, significantly reducing wall time as shown in Table 3. Shortcut Model policies trained in visual manipulation tasks achieved a comparable or slightly better success rate than DPPO, using as few as one step in Can and Square and four steps in Transport. DDIM policies in DPPO adopt five steps for these tasks.

**Flow Q Learning [38]**  Flow Q Learning (FQL) is an offline reinforcement learning algorithm designed for flow matching policies, which can also be used for offline-to-online fine-tuning.

Unlike our approach, FQL does not directly fine-tune a flow model with multiple denoising steps. Instead, it first trains a multi-step flow policy $\pi_\theta$ using the objective in Eq. (2). Then, it learns a Q function $Q_\phi$ from an offline RL dataset using one-step temporal difference (TD) learning, as described in [51].

$$\mathcal{L}(\phi) := \mathbb{E} \left[ \left( Q_\phi(s, a) - r - \gamma Q_{\bar{\phi}}(s', \pi_\omega(s', z)) \right)^2 \right] \tag{24}$$

At the same time, FQL distills a one-step policy $\pi_\omega$ from the pre-trained flow via minimizing the following loss:

$$\mathcal{L}(\omega) := \mathbb{E} \left[ -Q_\phi \left( s, \mu_\omega(s, z) \right) + \alpha \| \mu_\omega(s, z) - \mu_\theta(s, z) \|_2^2 \right] \tag{25}$$

The one-step policy will be deployed after FQL fine-tuning.

In the official FQL implementation, gradients flow to the multi-step policy during the distillation of the one-step policy, which can interfere with the pre-trained loss. As a result, we discover that even when we align the sample consumption during the offline RL phase, the multi-step FQL policy may struggle to match the reward achieved by pure behavior cloning pre-training.

In contrast, ReinFlow's method directly and stably fine-tunes a multi-step flow matching policy, offering richer representation capabilities than one-step distilled models in FQL. Empirically, ReinFlow outperforms FQL asymptotically and converges faster regarding wall-clock time, as shown in Figures 1 and 2. Additionally, ReinFlow does not rely on distillation. As a purely online RL algorithm, ReinFlow does not require labeled rewards for expert demonstrations.

**Other Methods**  DPPO [42] and FQL [38] provided an excellent introduction to a line of algorithms that fine-tune diffusion models with online RL and flow models with offline RL, including IDQL [23], QSM [40], DAWR [42], DIPO [55], DRWR [42], DQL [54], IFQL [38], etc. DPPO and FQL are generally superior to these methods regarding asymptotic reward, training stability, and wall time.

Although we have shown the advantage of ReinFlow over DPPO and FQL in Section 5, for completeness, we provide a brief comparison between the diffusion RL baselines with our method in a few representative continuous control tasks with the same set of hyperparameters indicated in [42]. Please refer to Appendix E for details.

## C Environment and Dataset Configuration

**Tasks.** Figures 8 and 9 demonstrate the locomotion and manipulation tasks considered in this work.

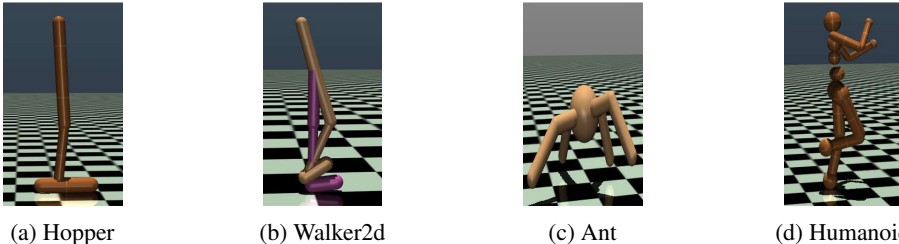

(a) Hopper      (b) Walker2d      (c) Ant      (d) Humanoid

Figure 8: Four OpenAI Gym locomotion Tasks: Hopper, Walker2d, Ant, and Humanoid.

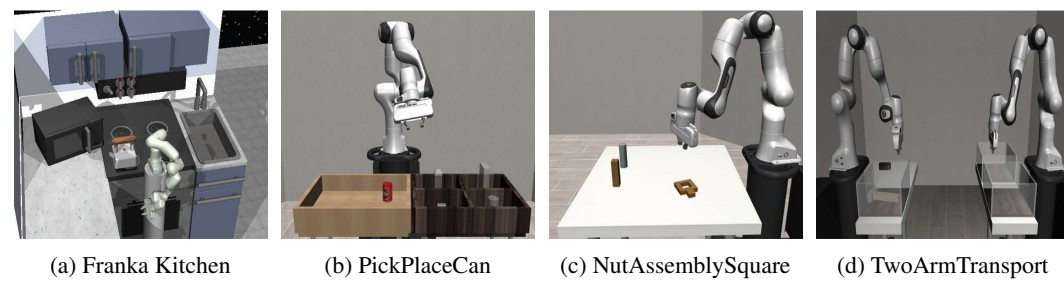

(a) Franka Kitchen      (b) PickPlaceCan      (c) NutAssemblySquare      (d) TwoArmTransport

Figure 9: Four manipulation tasks in state-input Franka Kitchen and pixel-input Robomimic environments.

**Environment Configuration.** We also list the environment configurations in different tasks in Table 1.

Table 1: Environment Configuration

| Environment | Task or Dataset | State Dim | Image Shape | Action Chunk | Max. Eps. Len.$H$ | Reward |
|---|---|---|---|---|---|---|
| OpenAI Gym | Hopper-v2 | 11 | - | $3 \times 4$ | 1000 | Dense |
| | Walker2d-v2 | 17 | - | $6 \times 4$ | 1000 | Dense |
| | Ant-v0 | 111 | - | $8 \times 4$ | 1000 | Dense |
| | Humanoid-v3 | 376 | - | $17 \times 4$ | 1000 | Dense |
| Franka Kitchen | Kitchen-Complete-v0 | 60 | - | $9 \times 4$ | 280 | Sparse |
| | Kitchen-Mixed-v0 | 60 | - | $9 \times 4$ | 280 | Sparse |
| | Kitchen-Partial-v0 | 60 | - | $9 \times 4$ | 280 | Sparse |
| Robomimic | Can | 9 | $[3,96,96] \times 1$ | $7 \times 4$ | 300 | Sparse |
| | Square | 9 | $[3,96,96] \times 1$ | $7 \times 4$ | 400 | Sparse |
| | Transport | 18 | $[3,96,96] \times 2$ | $14 \times 8$ | 800 | Sparse |

In Tab. 1, the "State Dim" indicates the dimension of proprioception inputs. "Action Chunk" expresses the dimension of a single action and the size of the action chunk. "Max.Eps.Len. $H$" indicates the maximum steps a robot can take in a single rollout. When the reward is "sparse, we award the agent a reward of $+1$ only upon task completion. Otherwise, the agent receives a 0 reward. The sparse reward is realistic, straightforward, and directly associated with the success rate of manipulation tasks. A "dense" reward system assigns the reward based on the robot's dynamics and kinematic properties. It is usually a floating number after each step of action execution. Dense reward systems need careful design. We adopt them primarily in sim2real training for legged locomotion tasks.

**The Robomimic Dataset Provided by DPPO [42]**    We remark that, in Robomimic environments, we adhere to the configuration of DPPO [42], which makes the shapes of the state vector smaller than the official default implementation in [36]. This processing simplifies robot learning and makes the pre-trained data provided by [42] smaller and simpler than Robomimic's official datasets. As acknowledged by the authors of DPPO, the datasets provided by [42] are of lower quality and/or quantity than the official release, so the pre-trained policy is trained on inferior data. Consequently, the pre-trained checkpoints of both DDPM and Shortcut policies have a lower success rate than the best-reported results in the literature [18].

We can train flow matching or diffusion policies with RoboMimic's default configuration and datasets, which will help us align with state-of-the-art results in the imitation learning literature. However, we may need to adopt larger neural networks and spend more time pre-training and fine-tuning.

**Datasets for OpenAI Gym Tasks**    To faithfully replicate the results of DPPO [42], we trained both DPPO and ReinFlow agents using the behavior cloning (BC) dataset provided by Ren et al. (2024) [42]. For consistency, we followed their dataset choices for all experiments in the Appendix that do not involve offline reinforcement learning (RL) algorithms, such as FQL.

However, the DPPO dataset for OpenAI Gym tasks lacks the offline rewards necessary to train an FQL agent, and the authors did not specify how they constructed their Gym dataset. We hypothesize that their BC datasets were derived and augmented from D4RL [19]. Therefore, in the experiments that involve FQL (described in Section 5.2 of the main text), we adopted D4RL offline RL datasets to train DPPO, FQL, and ReinFlow agents. We also remark that while we chose the "Ant-v2" environment in these experiments, we switched to "Ant-v0" in the sensitivity analysis in Section 4.3 and the Appendix.

The minor differences in OpenAI Gym Tasks mentioned above resulted in minimal discrepancy in reward curves.

# D    Implementation Details

## D.1    Model Architecture

For a fair comparison, we try to make the model architecture of flow matching policies align with diffusion policies proposed in DPPO [42] as much as possible.

**State-input Tasks.**    In state-input tasks (OpenAI Gym and Franka Kitchen), the velocity nets $v_\theta$ of 1-ReFlow policies are parameterized with Multi-layer Perceptrons (MLP) that receive action chunk, state, and time features. The time input $t \in (0, 1)$ is encoded by sinusoidal positional embedding [25] and linear projections, with a Mish [37] activation function in between. We implement Shortcut Models similarly, passing the time and inference step counts through the same sinusoidal positional embedding and concatenating. At the same time, we also encode the state input with a small MLP and add the state feature with the time-step features. The sizes of Shortcut Models are often slightly smaller than 1-ReFlow policies in the same task. Critic networks are also MLPs with the same or half the width.

**Pixel-input Tasks.**    In Robomimic, where the agent receives pixel and proprioception inputs, the actor and the critic adopt a single-layer Visual Transformer [15] with random shift augmentation as the visual encoder and compresses the proprioception information with a small MLP. We pass the features of the visuomotor condition, time embedding, and the raw action chunk through the actor's velocity head, which outputs a velocity estimate. The critic only receives features from time and condition.

**Noise Injection Net**    The noise injection network $\theta'$ shares the input features with the actor and outputs the standard deviation of the noise at each coordinate of the actions in the action chunk. The output of network $\theta'$ is passed through a $\text{Tanh}$ function coupled with affine transform, to ensure bounded output and smooth gradients. The upper and lower bounds of the noise standard deviation $\sigma_{\max}, \sigma_{\min}$ are a group of essential hyperparameters of ReinFlow, and we have provided an analysis in Section 6 to study how they influence exploration.

We discard $\sigma_{\theta'}$ after fine-tuning. Although a noise-injected process exists during RL, ReinFlow still returns a policy with an ODE inference path.

During the evaluation, we also did not inject noise into the flow policy; We observed that the reward is often higher than the noise-injected version, aligning with findings in classical RL literature [22].

**Parameter Scale**    We summarize the sizes of the actor, critic, and noise injection networks in Table 2, which shows that for state-input tasks, a negligible parameter increase ($< 6\%$) brought by the noise injection network returns promises a 135.36% net increase in reward and a 31.29% net increase in success rate.

For complex visual manipulation tasks beyond robomimic, since the noise network $\theta'$ shares the same representation backbone as the velocity network, the parameter increase should also remain minimal when scaling to larger models with more complex backbones, such as multi-layer Transformers, instead of the single-layer Transformer used in our experiments.

Designing an efficient noise injection network architecture that balances reward improvement and parameter count is an interesting problem, which we leave for future work.

Table 2: Model Parameter Counts

| Task | Model | Pre-trained Actor $\theta$/M | Noise Net $\theta'$/M | Fine-tuned Actor $\bar{\theta}$/M | Critic /M | Total /M | Noise/ Velocity |
|---|---|---|---|---|---|---|---|
| Hopper-v2 | 1-ReFlow | 0.55 | 0.01 | 0.56 | 0.13 | 0.69 | 1.22% |
| Walker2d-v2 | 1-ReFlow | 0.57 | 0.01 | 0.58 | 0.14 | 0.71 | 1.39% |
| Ant-v3 | 1-ReFlow | 0.62 | 0.01 | 0.64 | 0.16 | 0.80 | 2.31% |
| Humanoid-v3 | 1-ReFlow | 0.80 | 0.03 | 0.83 | 0.23 | 1.06 | 4.23% |
| Kitchen | Shortcut | 0.16 | 0.01 | 0.17 | 0.15 | 0.31 | 5.46% |
| Can | Shortcut | 1.01 | 0.15 | 1.16 | 0.59 | 1.74 | 14.59% |
| Square | Shortcut | 1.69 | 0.32 | 2.01 | 0.59 | 2.59 | 18.91% |
| Transport | Shortcut | 1.87 | 0.35 | 2.22 | 0.66 | 2.87 | 18.84% |

## D.2    Enhancing Training Stability

**Clipping Probability Ratio**    For stability reasons, we clip the log probability ratio in Algorithm 1 with $\epsilon = 0.01$ for state-input tasks and $\epsilon = 0.001$ for pixel-input tasks. This choice follows DPPO [42].

**Clipping Denoised Actions**    Although unnecessary for the pre-trained flow policies, we discover that during fine-tuning, it is beneficial to clip the denoised actions of a flow matching policy because this helps prevent the injected noise from interrupting the integration path too violently. After fine-tuning with ReinFlow, policies should also clip the denoised actions during inference; otherwise, performance may likely deteriorate.

**Critic Warmup**    Training the critic network for several iterations before updating the actor is crucial for stable training and rapid convergence, particularly for larger models with visual inputs. We refer to this phase as "critic warm-up". Since we use RL as a fine-tuning approach, the critic should output a reasonably large value (at least positive) before the policy gradient starts. An excessively small or even negative initial output misleads the actor into considering its current actions as excessively unsatisfying, resulting in rapidly degrading rewards and value function estimates during the fine-tuning process.

Empirically, we recommend initializing the critic and/or adjusting the critic warm-up iteration number according to the initial reward of the pre-trained policy, the rollout step number, and the discount factor.

**Critic Overfitting and Initialization**    An excessively long warm-up period may lead to overfitting of the critic network, especially in cases where the critic is significantly smaller than the pre-trained policy (Table 2). When the critic overfits, the policy gradient loss could oscillate violently during fine-tuning.

One possible solution to address critic overfitting is to incorporate regularizations into the critic's training procedure. Another method is to limit the warm-up iterations and initialize the critic's last fully connected layer with a positive bias. The bias can be estimated by the pre-trained success rate. With a positive initial output, the critic requires fewer warm-up iterations to output an appropriate value estimate without overfitting the pre-trained policy's distributions.

# E    Additional Experimental Results

**Random seeds**   Unless specified, we train all the RL algorithms with three seeds for all tasks except for Franka Kitchen with mixed or partial data, where we adopt two extra seeds, in the main experiment section 5. These tasks involve long-horizon multi-task planning, and we discover that DPPO and ReinFlow exhibit high variations across different runs.

The RL fine-tuning curves show the reward or success rate averaged over these seeds, with shading representing the mean ± standard deviation to indicate variability across runs.

**Rendering Backend**   During training and wall time testing, the MuJoCo graphics rendering backend (MUJOCO_GL) is set to Embedded System Graphics Library (EGL) to accelerate the computation with GPU. If users do not have EGL support and they switch to software rendering (osmesa), or if multiple threads are running together on the same group of CPU kernels or the same GPU, the compute time may be longer than that described in Table 3.

Table 3: Per Iteration Wall-clock Time

| Task | Algorithm | Single Iteration Time/second | | | Average |
|---|---|---|---|---|---|
| | | First seed | Second seed | Third seed | Mean $\pm$ Std |
| Hopper-v2 | ReinFlow-R | 11.598 | 11.704 | 11.843 | $11.715 \pm 0.123$ |
| | ReinFlow-S | 12.051 | 12.127 | 12.372 | $12.290 \pm 0.141$ |
| | DPPO | 99.502 | 99.616 | 98.021 | $99.046 \pm 0.890$ |
| | FQL | 4.373 | 4.366 | 4.515 | $4.418 \pm 0.084$ |
| Walker2d-v2 | ReinFlow-R | 11.861 | 11.446 | 11.382 | $11.563 \pm 0.260$ |
| | ReinFlow-S | 12.393 | 12.690 | 13.975 | $13.019 \pm 0.841$ |
| | DPPO | 101.151 | 106.125 | 98.470 | $101.915 \pm 3.884$ |
| | FQL | 5.248 | 4.597 | 5.207 | $5.017 \pm 0.365$ |
| Ant-v0 | ReinFlow-R | 17.210 | 17.685 | 17.524 | $17.473 \pm 0.242$ |
| | ReinFlow-S | 17.291 | 17.821 | 18.090 | $17.734 \pm 0.407$ |
| | DPPO | 102.362 | 104.632 | 99.042 | $102.012 \pm 2.811$ |
| | FQL | 5.242 | 4.950 | 5.3086 | $5.167 \pm 0.191$ |
| Humanoid-v3 | ReinFlow-R | 31.437 | 30.223 | 31.088 | $30.916 \pm 0.625$ |
| | ReinFlow-S | 30.499 | 30.058 | 31.029 | $30.529 \pm 0.486$ |
| | DPPO | 109.884 | 105.455 | 113.358 | $109.566 \pm 3.961$ |
| | FQL | 5.245 | 4.981 | 5.522 | $5.249 \pm 0.271$ |
| Franka Kitchen | ReinFlow-S | 26.655 | 26.328 | 26.628 | $26.537 \pm 0.182$ |
| | DPPO | 81.584 | 84.646 | 83.245 | $83.158 \pm 1.533$ |
| | FQL | 5.245 | 4.981 | 5.522 | $5.249 \pm 0.271$ |
| Can (image) | ReinFlow-S | 219.943 | 216.529 | 217.711 | $218.061 \pm 1.734$ |
| | DPPO | 310.974 | 307.811 | 308.014 | $308.933 \pm 1.771$ |
| Square (image) | ReinFlow-S | 313.457 | 312.3 | 313.862 | $313.206 \pm 0.811$ |
| | DPPO | 438.506 | 440.212 | 434.773 | $437.830 \pm 2.782$ |
| Transport (image) | ReinFlow-S | 554.196 | 557.712 | 559.006 | $558.359 \pm 0.915$ |
| | DPPO | 406.607 | 439.268 | 412.077 | $419.317 \pm 17.493$ |

**Wall-clock Time**   For a fair comparison, we measure all algorithms' wall-clock time in all the tasks (except Transport) on a single NVIDIA RTX 3090 GPU with EGL rendering. We evaluate the wall time of Robomimic Transport on two NVIDIA A100 GPUs with EGL rendering, as this task occupies significantly more memory than a single RTX 3090 GPU. The measurement is taken sequentially for three random seeds without interfering with other processes. We obtain the total wall-clock time for each algorithm by multiplying the average iteration time by the number of iterations. Table 3 provides a detailed record of these measurements, where the results are recorded in seconds and accurate to three decimal places.

Although FQL's per-iteration wall time is significantly shorter than other methods, this does not imply that it is more efficient overall: FQL's batch size is considerably smaller than that of PPO-based methods, including ReinFlow and DPPO, and its total training iterations are significantly larger than those of the other two methods; neither is FQL designed for parallel computing like DPPO and ReinFlow.

**Performance Increase**   We report the performance improvement after fine-tuning flow matching policies with ReinFlow across various simulation tasks in Table 4.

For locomotion tasks, we compute the reward increase ratio as follows:

$$\text{Locomotion Reward Net Increase Ratio} := \frac{\text{Fine-tuned Reward} - \text{Fine-tuned Reward}}{\text{Pre-trained Reward}} \tag{26}$$

For manipulation tasks, we compute the success rate increase:

$$\text{Manipulation Success Rate Net Increase} := \text{Fine-tuned Success Rate} - \text{Pre-trained Success Rate} \tag{27}$$

Policies fine-tuned with ReinFlow achieved an average episode reward net increase of $\mathbf{135.36\%}$ in OpenAI Gym locomotion tasks using D4RL datasets, with an average success rate increase of $\mathbf{31.29\%}$ in Franka Kitchen, $\mathbf{45.77\%}$ in Robomimic, and $\mathbf{40.34\%}$ in all manipulation tasks. The results are accurate to two decimal places.

**Sample Complexity in Gym Tasks**   Here, we also compare the sample complexity in OpenAI Gym tasks omitted in the main text. While FQL is more sample efficient in simpler tasks such as Hopper and Walker2d, it generally struggles to tackle more complex locomotion task, where it is asymptotically inferior to DPPO and ReinFlow.

**Comparison with Other Diffusion RL Methods**   We compare the diffusion RL baselines with our method in a few representative continuous control tasks. For these baselines, we adopt the same set of hyperparameters described in [42].

Figs 11 and 12 show that ReinFlow overall outperforms other methods concerning the stability w.r.t random seeds and asymptotic performance.

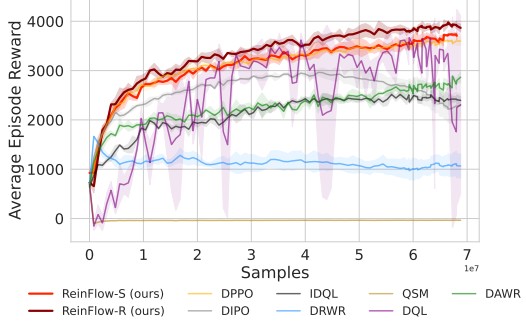

Figure 11: Fine-tuning locomotion task Ant-v0 with Diffusion RL baselines and ReinFlow.

Table 4: Performance Metrics for ReinFlow Across Tasks.

(a) Average Episode Reward in Locomotion Tasks.

| Task | Algorithm | Pre-trained Episode Reward | Fine-tuned Episode Reward | Reward Net Increase Ratio |
|---|---|---|---|---|
| Hopper-v2 | ReinFlow-R | 1431.80±27.57 | 3205.33±32.09 | 123.87% |
| | ReinFlow-S | 1528.34±14.91 | 3283.27±27.48 | 114.83% |
| Walker2d-v2 | ReinFlow-R | 2739.90±74.57 | 4108.57±51.77 | 49.95% |
| | ReinFlow-S | 2739.19±134.30 | 4254.87±56.56 | 55.33% |
| Ant-v2 | ReinFlow-R | 1230.54±8.18 | 4009.18±44.60 | 225.81% |
| | ReinFlow-S | 2088.06±79.34 | 4106.31±79.45 | 225.81% |
| Humanoid-v3 | ReinFlow-R | 1926.48±41.48 | 5076.12±37.47 | 163.49% |
| | ReinFlow-S | 2122.03±105.01 | 4748.55±70.71 | 123.77% |

(b) Average Success Rate in Manipulation Tasks.

| Environment and Task | Algorithm | Pre-trained Success Rate | Fine-tuned Success Rate | Success Rate Net Increase |
|---|---|---|---|---|
| Kitchen-complete | ReinFlow-S | 73.16±0.84% | 96.17±3.65% | 23.01% |
| Kitchen-mixed | ReinFlow-S | 48.37±0.78% | 74.63±0.36% | 26.26% |
| Kitchen-partial | ReinFlow-S | 40.00±0.28% | 84.59±12.38% | 44.59% |
| Can (image) | ReinFlow-R | 59.00±3.08% | 98.67±0.47% | 39.67% |
| | ReinFlow-S | 57.83±1.25% | 98.50±0.71% | 40.67% |
| Square (image) | ReinFlow-R | 25.00±1.47% | 74.83±0.24% | 49.83% |
| | ReinFlow-S | 34.50±1.22% | 74.67±2.66% | 40.17% |
| Transport (image) | ReinFlow-S | 30.17±2.46% | 88.67±4.40% | 58.50% |

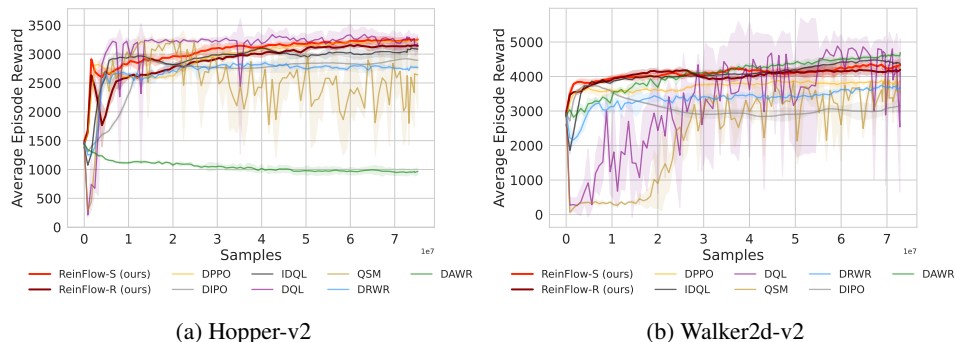

(a) Hopper-v2      (b) Walker2d-v2

Figure 12: Fine-tuning locomotion task Hopper-v2 and Walker2d-v2 with Diffusion RL baselines and ReinFlow.

**Changing the Behavior Cloning Dataset's Scale in Square**  Tab. 5 shows how the fine-tuned performance of ReinFlow is affected by the scale of the behavior cloning dataset in robomimic square. Fine-tuning the policy trained on 16 episodes failed due to an overly low initial success rate. We adopt the same hyperparameter set when fine-tuning policies trained on 64 and 100 episodes, restricting the

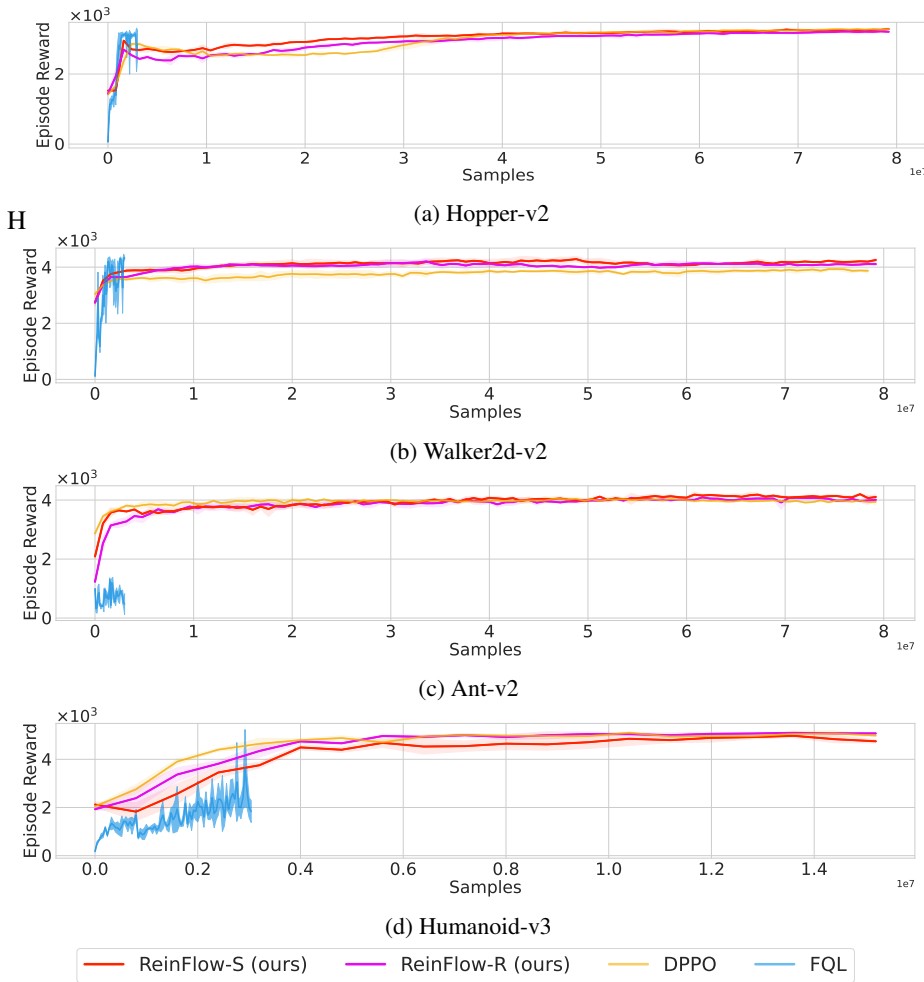

Figure 10: Sample efficiency results of state-based locomotion tasks in OpenAI Gym. For better visualization, we down-sampled FQL's data by five times in the first three tasks and three times in "Humanoid-v3". Although FQL is more sample-efficient than DPPO and ReinFlow in simpler tasks, it struggles to achieve high reward in more challenging locomotion tasks.

noise standard deviation to $\text{std} \in [0.08, 0.14]$ with entropy coefficient $0.01$. We slightly tuned down the noise to $\text{std} \in [0.06, 0.10]$ and removed entropy regularization, as we find it is more beneficial to limit exploration when the pre-trained policy performs poorly.

Table 5: Fine-tuned Success Rates in Square Across Different Pre-trained Episodes

| Pre-trained Episodes | Pre-trained Success Rate | Fine-tuned Success Rate | | | Average Fine-tuned Success Rate | |
|---|---|---|---|---|---|---|
| | | Fist seed | Second seed | Third seed | Mean | Std |
| 16 | 3.08 % | 0.00 % | 0.00 % | 0.00 % | 3.08 % | 0.00 % |
| 32 | 10.15% | 22.20% | 22.00% | 18.50% | 20.90% | 1.70 % |
| 64 | 27.67% | 65.50% | 62.20% | 56.20% | 61.30% | 3.85 % |
| 100 | 25.14% | 79.50% | 78.20% | 74.50% | 77.40% | 2.12 % |

**Changing the Number of Fine-tuned Denoising Steps**   Altering the fine-tuned denoising step number $K$ could affect ReinFlow's performance since the pre-trained policy has different rewards when evaluated at different steps.

Fig. 13 shows the difference when fine-tuning a Shortcut Policy in Franka Kitchen at $K = 1, 2$, and $4$ denoising steps.

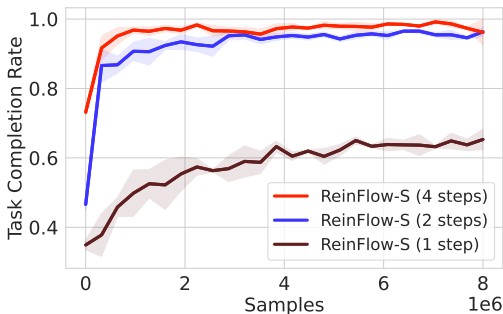

Figure 13: Fine-tuning Shortcut Policy in Kitchen-complete-v0 at Different Denoising Steps $1, 2$, and $4$.

In Fig. 13, all the run hyperparameters except $K$ and random seeds are the same. The pre-trained policy adopts complete demonstration data.

Increasing $K$ improves the reward at the beginning of fine-tuning. A better starting point shrinks the space of exploration and thus accelerates convergence. However, a longer denoising trajectory also consumes more simulation time. Fig. 13 and Fig. 4a also show that the success rate or episode reward quickly plateaus when scaling $K$.

In more challenging tasks such as visual manipulation, we also find that reducing the noise standard deviation is beneficial when we increase the number of denoising steps $K$, especially when the pre-trained policy has a low success rate.

## F    Discussion

In this section we provide additional discussion to help the readers better understand our design.

### F.1    Noise Injection versus Entropy Regularization

Both noise injection and entropy regularization serve to enhance exploration in reinforcement learning, but they play distinct and complementary roles in ReinFlow.

**Purpose and mechanism.** Noise injection is primarily used to replace the $\log \pi$ function, making the action distribution tractable. The added stochasticity naturally aids exploration as a byproduct. In contrast, entropy regularization provides an explicit mechanism to control exploration via its coefficient $\alpha$. The noise network enables ReinFlow to learn diverse actions by adapting noise intensity across denoising steps, while entropy regularization offers direct, tunable control over the exploration-exploitation trade-off. Importantly, ReinFlow remains mathematically valid even without the entropy term.

**Generalization beyond locomotion.** Entropy regularization is effective across various types of tasks, not only in locomotion domains. We successfully adopt it in the Robomimic visual manipulation task "square" (Table 9b, Appendix). Furthermore, in the "Franka Kitchen-complete" task—a state-input, sparse-reward manipulation environment—entropy regularization yields meaningful improvements in success rate. Ablation experiments averaged over 3 seeds show an increase from $96.17 \pm 3.65\%$ (without regularization of the entropy) to $99.00 \pm 0.75\%$ (with regularization of the entropy, $\alpha = 0.1$).

### F.2    Wall-time Efficiency

ReinFlow achieves faster wall-time performance for three key reasons:

(a) Fewer Denoising Steps: Flow models (e.g., ReFlows) require fewer steps than diffusion models (e.g., DDPM, DDIM) to reach high initial reward.

(b) Exact Probability Description in Few Steps: The probability of flow ODEs depends on the number of denoising steps. However, ReinFlow's expression remains exact even with very few steps, making it possible to take advantage of flow's property, enabling efficient fine-tuning without compromising reward.

(c) Simplified Optimization: We simplify DPPO's policy loss by not computing value/ advantage for denoised actions. This further reduces the computational cost.

### F.3  Additional Ablation Studies

We conducted additional ablation studies across various tasks during the rebuttal period, confirming that the observed trends are consistent.

**Noise conditioning across environments.** We compared the effect of different noise conditioning strategies in the Humanoid-v3 environment, conducting experiments with three random seeds. We evaluated conditioning only on state ($\sigma_{\theta'}(s)$) versus conditioning on both state and time ($\sigma_{\theta'}(s,t)$):

| Task | $\sigma_{\theta'}(s)$ | $\sigma_{\theta'}(s,t)$ |
|---|---|---|
| Humanoid | $4987.39 \pm 97.82$ | $5076.12 \pm 37.47$ |

These results demonstrate that conditioning the noise on both time and state yields higher rewards, consistent with the findings presented in Fig. 5 (page 9).

**Noise scale in sparse-reward manipulation.** We trained ReinFlow agents with different noise scales in the Franka Kitchen-complete task, with results averaged over three seeds:

| Noise std | Success Rate |
|---|---|
| 0.001 | $70.42\% \pm 3.21\%$ |
| 0.08 | $90.67\% \pm 14.87\%$ |
| 0.16 | $99.08\% \pm 1.01\%$ |

This experiment reveals that larger noise levels promote exploration. Together, these additional experiments demonstrate that the design choices and trends identified are robustly generalized across different types of environment, reward structures, and task complexities.

## G    Reproducing Our Findings

We list the key hyper-parameters and model architectures needed to reproduce the experiment results of ReinFlow and other baseline algorithms.

### G.1   Hyperparameters of ReinFlow

We adopt the same batch size suggested in DPPO [42] and follow their implementation to normalize the reward with time-reversed running variance.

It is essential to adjust the number of critic warmup iterations according to the performance of the pre-trained policy. A larger initial reward requires more warmup steps.

We clipped the denoised actions within $[-1, 1]$ to enhance training stability in implementation. Table 7 indicates this option with "clip intermediate actions". We also find it beneficial to slightly reduce the maximum noise standard deviation in the latter training course to postpone the reward decrease. $\cos(r_1, r_2)$ indicates a learning rate that decays in the rate of a cosine function from $r_1$ to $r_2$.

Table 6: ReinFlow's Shared Hyperparameters Across All Tasks

| Parameter | Value |
|---|---|
| critic loss coefficient | 0.50 |
| GAE lambda $\lambda$ | 0.95 |
| reward scale | 1.0 |
| reward normalization | True |
| actor optimizer | Adam [26] |
| actor learning rate weight decay | 0 |
| actor learning rate scheduler | CosineAnnealingWarmupRestart [35] |
| actor learning rate cycle steps | 100 |
| critic optimizer | Adam |
| critic learning rate scheduler | CosineAnnealingWarmupRestart |
| critic scheduler warmup | 10 |
| critic learning rate cycle steps | 100 |

Table 7: ReinFlow's Hyperparameters in OpenAI Gym Locomotion Tasks

(a) Shared Hyperparameters Across OpenAI Gym Tasks

| Parameter | Value |
|---|---|
| critic learning rate weight decay | 1e-5 |
| number of parallel environments | 40 |
| reward discount factor $\gamma$ | 0.99 |
| action chunking size | 4 |
| condition stacking number | 1 |
| batch size | 50k |
| maximum episode steps | 1000 |
| number of rollout steps | 500 |
| update epochs | 5 |
| number of training iterations | 1000 |
| number of denoising steps | 4 |
| clipping ratio $\epsilon$ | 0.01 |
| clip intermediate actions | True |
| target KL divergence | 1.0 |
| noise std upper bound hold for | 35% of total iteration |
| noise std upper bound decay to | $0.3 \times \sigma_{\min} + 0.7 \times \sigma_{\max}$ |
| entropy coefficient $\alpha$ | 0.03 |
| BC loss ($W_2$ regularization) coefficient $\beta$ | 0.00 |

(b) Task-Specific Hyperparameters in OpenAI Gym Environment

| Parameter | Hopper-v2 | Walker2d-v2 | Ant-v0 Humanoid-v3 |
|---|---|---|---|
| minimum noise std $\sigma_{\min}$ | 0.10 | 0.10 | 0.08 |
| maximum noise std $\sigma_{\max}$ | 0.24 | 0.24 | 0.16 |
| critic warmup iters | 0 | 5 | 0 |
| actor learning rate | cos(4.5e-5, 2.0e-5) | cos(4.5e-4, 4.0e-4) | cos(4.5e-5, 2.0e-5) |
| critic learning rate | cos(6.5e-4, 3.0e-4) | cos(4.0e-3, 4.0e-3) | cos(6.5e-4, 3.0e-4) |

Table 8: ReinFlow's Hyperparameters in Franka Kitchen State-input Manipulation Tasks

| Parameter | Value |
|---|---|
| critic learning rate weight decay | 1e-5 |
| number of parallel environments | 40 |
| reward discount factor $\gamma$ | 0.99 |
| action chunking size | 4 |
| condition stacking number | 1 |
| batch size | 5600 |
| maximum episode steps | 280 |
| number of rollout steps | 200 |
| update epochs | 10 |
| number of training iterations | 301 |
| number of denoising steps | 4 |
| clipping ratio $\epsilon$ | 0.01 |
| clip intermediate actions | True |
| minimum noise std $\sigma_{\min}$ | 0.05 |
| maximum noise std $\sigma_{\max}$ | 0.12 |
| noise std upper bound hold for | 100% of total iteration |
| noise std upper bound decay to | $\sigma_{\max}$ |
| entropy regularization coefficient $\alpha$ | 0.00 |
| BC loss ($W_2$ regularization) coefficient $\beta$ | 0.00 |

Table 9: ReinFlow's Hyperparameters in Robomimic Visual Manipulation Tasks

(a) Shared Hyperparameters Across Robomimic Tasks

| Parameter | Value |
|---|---|
| critic learning rate weight decay | 0 |
| number of parallel environments | 50 |
| reward discount factor $\gamma$ | 0.999 |
| condition stacking number | 1 |
| pixel input shape | [3, 96, 96] |
| image augmentation | RandomShift (padding=4) |
| gradient accumulation steps | 15 |
| batch size | 500 |
| update epochs | 10 |
| clipping ratio $\epsilon$ | 0.001 |
| clip intermediate actions | True |
| target KL divergence | 1e-2 |
| noise std upper bound hold for | 100% of total iteration |
| noise std upper bound decay to | $\sigma_{\max}$ |
| BC loss ($W_2$ regularization) coefficient $\beta$ | 0.00 |

(b) Task-Specific Hyperparameters in Robomimic Environment

| Parameter | PickPlaceCan | NutAssemblySquare | TwoArmTransport |
|---|---|---|---|
| minimum noise std $\sigma_{\min}$ | 0.08 | 0.08 | 0.05 |
| maximum noise std $\sigma_{\max}$ | 0.14 | 0.14 | 0.10 |
| number of denoising steps | 1 | 1 | 4 |
| entropy coefficient $\alpha$ | 0.00 | 0.01 | 0.00 |
| critic warmup iters | 2 | 2 | 5 |
| critic output layer bias | 0.0 | 0.0 | 4.0 |
| actor learning rate warmup | 10 | 25 | 10 |
| actor learning rate | cos(2.0e-5, 1.0e-5) | cos(3.5e-6, 3.5e-6) | cos(3.5e-6, 3.5e-6) |
| critic learning rate | cos(6.5e-4, 3.0e-4) | cos(4.5e-4, 3.0e-4) | cos(3.2e-4, 3.0e-4) |
| action chunking size | 4 | 4 | 8 |
| number of cameras | 1 | 1 | 2 |
| maximum episode steps | 300 | 400 | 800 |
| number of rollout steps | 300 | 400 | 400 |
| number of training iterations | 150 | 300 | 200 |

## G.2 Hyperparameters of DPPO

We strictly follow the hyperparameter setting of the official implementation of DPPO. Please refer to Section E.10 of [42] for details.

The only changes we make are incorporating seeds 509 and 2025 for furniture tasks and elongating the rollout steps from 70 to 200 (while we still keep the maximum episode length as 280 in the environment configuration). DPPO and ReinFlow exhibit higher variability in Franka kitchen tasks across random seeds and even different runs, so we adopt five random seeds. We also find that a longer sampling trajectory helps the DPPO agent discover optimal strategies. For this reason, DPPO achieves a higher task completion rate than that reported by DPPO's original paper.

## G.3 Hyperparameters of FQL

We list the hyperparameters adopted by FQL in Table 10. We set the number of offline pre-training steps such that the total sample consumption during the offline phase is no less than the pre-trained consumption of DPPO or ReinFlow in Gym (D4RL) and Franka Kitchen tasks.

As described in [38], the temperature coefficient, $\alpha_{\mathrm{FQL}}$, is the most important hyperparametero f FQL. We followed the instructions of the original paper and scanned $\alpha_{\mathrm{FQL}}$ in $[0.03, 0.1, 0.3, 1, 3, 10]$ to obtain a proper value for the temperature in Hopper-v2. and adopted the same value for all state-input tasks.

Table 10: Hyperparameters for FQL in Gym and Franka Kitchen Tasks

(a) Shared Hyperparameters for FQL

| Parameter | Value |
|---|---|
| number of denoising steps of the base policy $\pi_\theta$ | 4 |
| number of steps for evaluation | 500 |
| number of evaluation episodes | 10 |
| reward discount factor $\gamma$ | 0.99 |
| actor learning rate | $1e-4$ |
| actor weight decay | 0 |
| actor learning rate cycle steps | 1000 |
| actor scheduler warmup | 10 |
| critic learning rate | $3e-4$ |
| critic learning rate weight decay | 0 |
| critic scheduler warmup | 10 |
| batch size | 256 |
| target EMA rate | 0.005 |
| reward scale | 1.0 |
| buffer size | $1e6$ |
| Behavior cloning coefficient | 3.0 |
| actor update repeat | 1 |
| online steps | 569,936 |

(b) Task-Specific Hyperparameters in Robomimic Environment

| Parameter | Gym and Kitchen-complete-v0 | Kitchen-mixed-v0 and Kitchen-partial-v0 |
|---|---|---|
| offline steps | 200,000 | 600,000 |

