# OpenReview forum: "ReinFlow: Fine-tuning Flow Matching Policy with Online Reinforcement Learning"
_NeurIPS.cc/2025/Conference — NeurIPS 2025 poster_

### Official Review · Reviewer_aYCb · 2025-06-04

**Clarity:** 3
**Significance:** 2
**Originality:** 2
**Rating:** 4
**Confidence:** 4

**Summary:**

The authors propose ReinFlow, a method to fine-tune flow-matching policies for online RL. The main difficulty of fine-tuning flow policies with PPO-like online RL algorithms is in computing the log likelihood of actions. To deal with this, the authors convert the flow ODE into an SDE with noise injected at each step. The strength of the noises is controlled by another learnable network trained in an end-to-end manner. They show that ReinFlow outperforms DPPO and FQL on various robotics domains.

**Questions:**

* What's the disadvantage of using Eq. (4)? It seems this is the most straightforward way to fine-tune a flow policy with online RL. Have the authors empirically compared ReinFlow to Eq. (4) with (say) 10 flow steps?

**Ethical Concerns:**

["NO or VERY MINOR ethics concerns only"]

**Final Justification:**

My initial concerns about statistical significance have been resolved, and I see the relevance of this paper in that it proposes one of the first approaches that fine-tune a flow policy with policy gradients. That said, I'm still not convinced by the significance of their empirical results, especially in comparison to previous baselines (DPPO, SAC, etc.).

Given the above, I've adjusted my score to 4 (borderline accept), but I wouldn't be opposed to rejection either.

**Limitations:**

Limitations are not explicitly discussed, even though the authors claim in the checklist that they are discussed in the last section.

**Paper Formatting Concerns:**

I don't have paper formatting concerns.

**Quality:**

2

**Strengths And Weaknesses:**

Strengths
* The method is straightforward and reasonable.
* The paper is easy to understand.
* The authors evaluate ReinFlow across diverse settings (including pixel-based tasks).
* The authors provide quite extensive ablation studies about ReinFlow's different design choices.

Weaknesses
* The performance improvement over DPPO seems marginal in general. While the authors show that ReinFlow is better than DPPO in terms of *wall-clock* time, ReinFlow and DPPO achieve similar performance in terms of environment steps (Fig. 10 in the Appendix). Also, are there any reasons why ReinFlow should be faster than DPPO *in principle*, given they are both based on the same PPO framework with a similar iterative policy training scheme? Moreover, vanilla SAC achieves significantly better sample efficiency on all these Gym MuJoCo tasks (see the original SAC paper).
* Only two flow/diffusion baselines are considered in the main text. Since FQL is essentially an offline RL algorithm, the comparison is mainly done only against DPPO. While the authors compare with more baselines in Fig. 11 and 12 in the Appendix, again, the improvement is marginal, and vanilla SAC seems to outperform all of the baselines.
* The authors use only 3 seeds in most plots. In the Appendix, the authors mention using seeds {0, 42, 3407} (and additionally {509, 2025} for some Kitchen experiments). This seems quite arbitrary; it'd have been better if the authors had used more systematically generated numbers so that they didn't look cherry-picked.


Minor comments
* L47: "We propose ReinFlow, the first online RL algorithm to stably fine-tune a family of flow matching policies" -- I think this is an overclaim given the presence of Flow-GRPO.
* L127: Infact -> In fact
* L142: "the generated sigma algebra of $a_t$ and $t$" -- While random variables do generate sigma algebras, $a_t$ and $t$ seem deterministic variables here, technically.
* I'd highly recommend using consistent notations to denote times in flow matching (e.g., subscripts in Eq. (5) vs. superscripts in Eq. (6)).
* L157: Missing reference.
* L202: Use \citet.

---

> ### Author Rebuttal · Authors · 2025-07-29
>
> **General response to Reviewer aYCb**
>
> We thank your detailed review and the positive comment on the clarity of our presentation, the soundness of our algorithm design, and the comprehensiveness of our empirical results!
>
> * **The Reviewer’s Concern.**
> The reviewer raises concerns about ReinFlow's success rate relative to SAC and DPPO. However, _these methods serve different purposes_: SAC trains Gaussian policies from scratch, and DPPO fine-tunes Diffusion Policies. In contrast, ReinFlow is designed _as an online RL framework for fine-tuning flow matching policies obtained via imitation learning_—a growing focus in robot learning [1,3,4].
>
> * **Intent and Scope of Our Contribution.**
> While SAC and DPPO achieve near-optimal success rates in several tasks, focusing solely on this metric overlooks our contributions: a) ReinFlow improves performance _over imitation learning baseline_, with significant wall-time efficiency.
> b) It is explicitly designed for a broad class of _flow_ policies, an area not addressed by SAC or DPPO.
>
> ****
>
> **1. ReinFlow and SAC**
> > "Vanilla SAC achieves significantly better sample efficiency on Gym MuJoCo tasks (see the original SAC paper)."
>
> SAC is effective in simple MuJoCo tasks, but this does not diminish our contributions. In fact, its limitations in more complex domains highlight ReinFlow’s broader applicability.
>
> * **Different design goals**. SAC trains _Gaussian_ policies _from scratch_ [5], which often lacks the expressiveness needed in complex tasks [2]. ReinFlow instead fine-tunes _flow matching_ policies _following imitation learning_—a important direction gaining significant traction [1,3,4].
>
> * **ReinFlow's flexible implementation**. While ReinFlow currently builds on PPO, it can, in principle, incorporate SAC-style updates via Theorem 4.1 (lines 158–162), potentially improving sample efficiency (lines 212–214).
>
> * **ReinFlow excels in complex tasks, where SAC fails**
>   - Although effective in dense-reward MuJoCo tasks, _SAC performs poorly_ in sparse-reward, long-horizon manipulation tasks, such as Franka Kitchen ([7], page 8, Fig. 5) and Robomimic ([6], page 17).
>   - DPPO [2] explicitly reports it is "...the first RL algorithm to solve Transport with >50% success rates" (page 11), implying SAC's limitations.
>   - In contrast, _ReinFlow_ achieves >70% success rates across all tested sparse-reward manipulation tasks (Appendix, page 30), _demonstrating robust performance where SAC fails_.
>
> * **ReinFlow matches or outperforms SAC’s flow variant, FQL.**
> FQL is a SAC-based method for flow policies [1], and ReinFlow matches or exceeds its reward in MuJoCo and Franka Kitchen , while achieving superior wall-time efficiency (Section 5.2, page 7).
>
> We thank the reviewer for highlighting the importance of sample efficiency. We plan to integrate SAC-style off-policy updates for ReinFlow in future work.
>
> ****
>
> **2. ReinFlow and DPPO**
>
> > 2.1 Only two flow/diffusion baselines are considered in the main text...The performance improvement over DPPO seems marginal in general.
>
> * **On the choice of baselines**: We emphasize FQL and DPPO in the main text because they represent state-of-the-art methods for flow and diffusion models at the time of submission, respectively [1,2]. Moreover, six additional baselines are included in the Appendix (page 31).
>
> * **Success Rate and the Intention of Our Work**.
>   - **Our goal** is not to significantly outperform DPPO in success rate, as it is nearly-optimal in many tasks, especially in Robomimic [2]. Rather, we aim to introduce the first online RL framework tailored for flow policies, and improve the success rate when data is suboptimal (lines 22-34, page 1)—an area that lacked an equivalent to DPPO and is gaining significant attention [1,3,4].
>   - **Our contribution**: Unlike diffusion which rely on SDEs, flows based on ODEs and require different treatment. ReinFlow fills this gap, matching or beating DPPO while offering substantial wall-time reduction (Section 5.2, page 7). In this sense, ReinFlow serves flow policies as effectively as DPPO does for diffusion policies, if not better.
>
> > 2.2 ReinFlow and DPPO achieve similar performance in terms of environment steps.
>
> We focus on wall-time efficiency, not sample efficiency, given the simulated nature of our experiments (page 6). While improving sample efficiency is important—especially for real-world RL—we leave this as promising future work.
>
> > 2.3 Why ReinFlow should be faster than DPPO in _principle_ ?
>
> ReinFlow achieves faster wall-time performance for three key reasons:
>  * (a)  **Fewer Denoising Steps:**  Flow models (e.g., ReFlows) require fewer steps than diffusion models (e.g., DDPM, DDIM) to reach high initial reward. ([8], Fig. 5.4; [9], page 8).
> *  (b)  **Exact Probability Description at Few Steps:**  The probability of flow ODEs depends on the denoising step number. However, ReinFlow's expression remains exact even with very few steps (lines 138–147), making it possible to take advantage of flow's property, enabling efficient fine-tuning without compromising reward.
> *  (c)  **Simplified Optimization:**  We simplify DPPO’s policy loss by not computing value/ advantage for denoised actions (Appendix C, page 25). This further reduces computational cost.
>
> We appreciate your question and plan to better explain wall-time efficiency in our revision.
>
> ****
>
> **3. Random Seeds**
> > ...it'd have been better if ...used more systematically generated numbers...
>
> We took your advice seriously and ran additional experiments using seeds 1, 2, 3, 4, 5.
>
> Due to time limit, we train Rectified Flows in locomotion and Shortcut Models in manipulation.
>
> |Task|Original Seeds|New seeds|
> |----|----|---|
> |Hopper|3205.33 ± 32.09|3197.33 ± 60.95|
> |Walker2d|4108.57 ± 51.77|4171.81 ± 141.88|
> |Ant|4009.18 ± 44.60|3884.52 ± 161.51|
> |Humanoid|5076.12 ± 37.47|5007.35 ± 88.25|
> |Kitchen-complete|96.17% ± 3.65%|98.25% ± 0.66%|
> |Kitchen-mixed|74.63% ± 0.36%|74.85% ± 0.14%|
> |Kitchen-partial|84.59% ± 12.38%|70.60% ± 19.73%|
> |Can|98.50% ± 0.71%|98.70% ± 1.35%|
> |Square|74.67% ± 2.66%|71.10% ± 4.64%|
> |Transport|88.67% ± 4.40%|85.75% ± 2.90%|
>
> While the differences are not substantial, we will follow your advice and update the experimental section accordingly.
>
> ****
>
> **4. Minor comments**
>
> > L47: "ReinFlow, the first online RL algorithm to stably fine-tune a family of flow matching policies" -- this is an overclaim given Flow-GRPO"
>
> Flow-GRPO is a concurrent work and we clarified the distinctions in Section 2 (lines 88–90, page 3). To the best of our knowledge, our claim is valid at the time of submission.
>
> * Flow-GRPO does not train flow-matching *policies* for robotic control. It targets flow-matching models for image generation ([11] , Section 5, page 5).
> * Flow-GRPO's mathematical derivations are specific to Rectified Flows (Section 3, page 3; Eqs. 26–29, page 17 of the Appendix). In contrast, ReinFlow applies to broader flow model families, including Shortcut Models [10] (Section 5.2, page 7, Theorem 4.1 page 5).
>
> > "L142: generated sigma algebra."
>
> We refer to the "generated sigma algebra" because in ReinFlow, the denoised actions are stochastic due to the injection of Gaussian noise (Algorithm 1, page 4).
>
> > "What's the disadvantage of using Eq. (4)? "
>
> Thank you for raising this point!  Eq. (4) is accurate only when the step size is very small (lines 132–134, page 4)—contrary to our goal of high inference speed. Moreover, Eq. (4) introduces Monte Carlo noise, whereas our method offers exact estimation of the action log probability (lines 138–147) at arbitrarily few steps.
>
> We observed that using Eq. (4) with deterministic policy gradient methods (e.g., TD3) led to unstable training, while ReinFlow produced stable and reliable rewards (Section 5.2, page 7). We will emphasize this in future revisions.
>
> > Limitations are not explicitly discussed.
>
> We do acknowledge the limitations of our work in Section 7, page 9.
>
> > I'd highly recommend using consistent notations to denote times in flow matching. L157: Missing reference. L202: Use \citet. L127: Infact -> In fact
>
> Thank you! We will fix the typos in the revised manuscript.
>
> ****
>
> **Final Remark**
>
> The reviewer is concerned with the comparison to SAC in locomotion. While SAC is effective on simple benchmark, ReinFlow excels in **more complex tasks** and is designed for **a new purpose**--**to improve flow policy after imitation learning**. We respectfully ask the reviewer to reconsider the assessment of our work's quality and significance, and we're happy to clarify any questions.
>
> ****
>
> **References**
>
> [1] Park, Seohong, et al. "Flow q-learning." arXiv:2502.02538 (2025).
>
> [2] Ren, Allen Z., et al. "Diffusion policy policy optimization." arXiv:2409.00588 (2024).
>
> [3] Braun, Max, et al. "Riemannian flow matching policy for robot motion learning." 2024 IEEE/RSJ International Conference on Intelligent Robots and Systems (IROS). IEEE, 2024.
>
> [4] Black, Kevin, et al. "$\pi_0 $: A Vision-Language-Action Flow Model for General Robot Control." arXiv:2410.24164 (2024).
>
> [5] Haarnoja, Tuomas, et al. "Soft actor-critic: Off-policy maximum entropy deep reinforcement learning with a stochastic actor." International conference on machine learning. Pmlr, 2018.
>
> [6] Zhu, Yuke, et al. "robosuite: A modular simulation framework and benchmark for robot learning." arXiv:2009.12293 (2020).
>
> [7] Zhang, Jesse, et al. "EXTRACT: Efficient Policy Learning by Extracting Transferable Robot Skills from Offline Data." arXiv:2406.17768 (2024).
>
> [8] Zheng, Qinqing, et al. "Guided flows for generative modeling and decision making." arXiv:2311.13443 (2023).
>
> [9] Zhang, Fan, and Michael Gienger. "Affordance-based robot manipulation with flow matching." arXiv:2409.01083 (2024).
>
> [10] Liu, Jie, et al. "Flow-grpo: Training flow matching models via online rl." arXiv:2505.05470 (2025).
>
> [11] Frans, Kevin, et al. "One step diffusion via shortcut models." arXiv:2410.12557 (2024).

---

> > ### Comment · Reviewer_aYCb · 2025-07-31
> >
> > Thanks for the response.
> >
> > While I still think the results of ReinFlow are less convincing, given that it underperforms SAC on Gym MuJoCo tasks and is only marginally better than DPPO on Kitchen/Robomimic tasks, I think this paper does have value in that it proposes one of the first approaches that fine-tune a flow policy with policy gradients. I also appreciate the new results with more seeds. I'd like to strongly encourage the authors to update the paper results in the final version.
> >
> > (Minor) The authors mentioned "We do acknowledge the limitations of our work in Section 7, page 9." in the rebuttal, but I still couldn't find an explicit discussion about limitations in Section 7. Perhaps you meant "It is an exciting direction to implement ReinFlow with various other policy gradient methods and scaling up the fine-tuned policy to large vision-language-action models, which we leave to future work"? I don't think this counts as a limitation and would like to encourage the authors to discuss limitations in the next revision.
> >
> > Given the above, I'd like to adjust my score to 4 (borderline accept), but I wouldn't be opposed to rejection either.

---

> > > ### Author Response · Authors · 2025-08-01
> > > **Reply to Reviewer aYCb**
> > >
> > > We appreciate your prompt response!
> > >
> > > We will follow your valuable advice and update our paper with the new runs in our experiment. In our next revision, we will also add a separate section to explicitly discuss the limitations. Thank you for your thoughtful review!

---

### Official Review · Reviewer_S8sj · 2025-06-30

**Clarity:** 3
**Significance:** 3
**Originality:** 3
**Rating:** 4
**Confidence:** 2

**Summary:**

The document introduces ReinFlow, a novel online reinforcement learning framework designed to fine-tune flow matching policies for continuous robotic control. Unlike previous methods, ReinFlow addresses challenges in likelihood computation and exploration for flow models by injecting learnable noise, converting the deterministic flow into a discrete-time Markov process. This approach significantly improves the success rate and reward of pre-trained policies in various locomotion and manipulation tasks, including those with visual input and sparse rewards. The paper demonstrates ReinFlow's better efficiency compared to state-of-the-art diffusion RL methods and examines key factors influencing its performance, such as noise levels and regularization techniques.

**Questions:**

For the results in Figure 1. Is the ReinFlow baseline based on similar implementations as the baselines? It would be nice to know how the wall-clock savings can be broken down as implementation efficiency + algorithmic efficiency.
For evaluation, it looks like you keep most hyperaparameters of the baseline (DPPO) intact. That implies that DPPO is using a lot more denoising steps which is computationally expensive. I would like to see some evaluation between ReinFlow and DPPO with comparable denoising steps in terms of sample complexity and wall-clock time. That would allow a better assessment of the algorithmic improvements made in this work.

I am willing to increase my score if the authors can address my comments above.

Nit:
* Line 269 Figure 6 - you are missing the super caption for this figure. 6(a) the color encoding is repeated for std = 0.160 and 0.040. Also, the color coding is difficult to see.
* Line 1011 "asymptotically inferior” In the evaluation of FQL on sample complexity, it looks like it is not run with the same number of total steps.
* 5.2 Performance evaluation x-axis as number of frames instead of wall-clock time. What is the confidence interval reported?
* Line 157 missing reference to Appendix.
* DPPO evaluated on halfcheetah. Lift - state (https://arxiv.org/pdf/2409.00588). It would be great if you can also include it in your evaluation.
* Line 127 “infact”

**Ethical Concerns:**

["NO or VERY MINOR ethics concerns only"]

**Final Justification:**

The authors have addressed my points.

**Limitations:**

Yes.

**Paper Formatting Concerns:**

No.

**Quality:**

3

**Strengths And Weaknesses:**

The biggest strength of the proposed method is the improvement in wall-clock efficiency - Reinflow can achieve comparable performance as DPPO at a fraction of the wall clock time. There are also improvements in asymptotic performance in various tasks.
There are good evaluation and ablation on numerous environments which help understand the effect of ReinFlow’s hyperparameters.

I have some questions regarding some experimental evaluation which I have detailed in the questions section.

---

> ### Author Rebuttal · Authors · 2025-07-30
>
> **General Response to Reviewer S8sj**
>
> Thank you for your positive and constructive feedback!
>
> We appreciate your recognition of the originality and novelty of our approach, how we address the challenge of likelihood computation and ReinFlow's significant improvement in success rate and wall-time efficiency. We are also grateful for your comments on the thoroughness of our sensitivity analysis.
>
> ****
>
> **1. Mechanism Behind Wall-Time Reduction**
>
> > ...It would be nice to know how the wall-clock savings can be broken down as implementation efficiency + algorithmic efficiency...I would like to see some evaluation between ReinFlow and DPPO with comparable denoising steps in terms of sample complexity and wall-clock time. That would allow a better assessment of the algorithmic improvements made in this work.
>
> > I am willing to increase my score if the authors can address my comments above.
>
> Thank you for raising this important point. We address it below:
>
> * **Implementation Consistency**.
> We use the same implementation as DPPO wherever possible, including training loops, buffer structures, and shared hyperparameters. DPPO's hyperparameters remain unchanged from its official release, and our submitted code in the supplementary materials reflects this.
>
> * **Why is ReinFlow faster?** The observed wall-time savings stem primarily from algorithmic efficiency.
>   - a). **Fewer Denoising Steps**: Flow policies such as Rectified Flow need fewer integration steps than DDPM/DDIM to achieve similar precision, as evidenced by [1,2,3].
>   - b). **Exact Likelihood**: ReinFlow precisely calculates the flow policy’s probability at any steps, enabling training with as few as four or even one step (Section 4.1, page 3).
>   - c). **Stabilized Fine-Tuning**: By controlling the magnitude of noise and denoised actions (Appendix, lines 845–852, page 22), ReinFlow avoids abrupt policy update during fine-tuning, ensuring a stable increase in success rates
>   - d). **Simplified Computation in Optimization**: Unlike DPPO, ReinFlow avoids computing the value and advantage functions at each denoised step, saving optimization time (Appendix C, page 25).
>
> * **If DPPO uses the same steps as ReinFlow, what will happen?**
>
> We thank the reviewer for the helpful feedback. To address the question directly, we ran both DPPO and ReinFlow with 4 denoising steps during evaluation and RL training. Experiments were conducted on a single NVIDIA RTX 3090 GPU with EGL backend and averaged over seeds 1, 2, and 3.
>
>   - **Inference Time per Action (ms)**
> | Task              | DPPO         | ReinFlow         |
> |-------------------|--------------|------------------|
> | Hopper            | 4.25 ± 0.16  | 3.71 ± 0.21   |
> | Walker            | 4.47 ± 0.25  | 3.90 ± 0.19   |
> | Kitchen-complete  | 7.22 ± 0.08  | 6.67 ± 0.06 |
>
>   - **RL Reward / Success Rate**
> | Task              | DPPO                | ReinFlow             |
> |-------------------|---------------------|-----------------------|
> | Hopper            | 2689.54 ± 35.47     | 3197.33 ± 60.95   |
> | Walker            | 3688.90 ± 60.83     | 4171.81 ± 141.88  |
> | Kitchen-complete  | 5.83% ± 0.95%       | 98.25% ± 0.66%    |
>
>   - **RL Wall-Time per Iteration (s)**
> | Task              | DPPO         | ReinFlow         |
> |-------------------|--------------|------------------|
> | Hopper |  43.95 ± 1.41 | 14.17 ± 0.35 |
> |Walker| 45.03 ± 0.59 | 14.46 ± 0.21 |
> | Kitchen-complete  | 41.71 ± 0.52 | 26.56 ± 0.38  |
>
>   - **Conclusion:**
>     - During inference, DPPO and ReinFlow have similar per-action time, with ReinFlow slightly faster due to the use of an ODE solver rather than an SDE solver.
>     - During RL, ReinFlow reduces wall-time per iteration by accelerating both rollout and optimization, resulting in faster training.
>     - With the same number of denoising steps, DPPO starts with lower rewards and continues to lag behind, while ReinFlow achieves significantly better performance, combining fast inference with high reward.  These trends align with prior findings ([1], Fig. 5.4 and 5.5, p. 9).
>
> We appreciate the reviewer’s suggestion and will include further discussion on ReinFlow’s wall-time efficiency in future versions.
>
> ****
>
> **2. Confidence interval**
>
> > What is the confidence interval reported?
>
> Thank you for this question! As noted in Appendix (Page 28, Lines 981–982), we report **one-sigma error bars**, with shaded areas indicating one standard deviation from the mean. We will clarify this in the main text to improve its visibility.
>
> ****
>
> **3. Additional experiments**
>
> >   DPPO evaluated on halfcheetah. Lift - state... It would be great if you can also include it in your evaluation.
>
> We thank the reviewer for the suggestions to benchmark in additional state-based tasks. In Section 5.2 (page 7), we have chosen a diverse set of tasks--both state and visual input--with varying difficulty for a representative evaluation.
>
> While time constraints prevent us from adding all suggested tasks during the rebuttal phase, we agree with the value of this suggestion and plan to incorporate them in future work or a camera-ready revision.
>
> ****
>
> **4. Typos and Formatting**
>
> > Line 269 Figure 6 - you are missing the super caption for this figure. 6(a) the color encoding is repeated for std = 0.160 and 0.040. Also, the color coding is difficult to see.
> > Line 127 “in fact”
> > 5.2 Performance evaluation x-axis as number of frames instead of wall-clock time.
> > Line 157 missing reference to Appendix.
> > Line 1011 "asymptotically inferior” In the evaluation of FQL on sample complexity, it looks like it is not run with the same number of total steps.
>
> Thank you for the thorough review! We will fix these issues in our subsequent revisions.
>
> ****
>
> **Final Remark**
>
> We hope we have addressed your concerns and we appreciate it if you kindly consider improving our rating. Please also let us know if you have any remaining questions!
>
> ****
>
> **References**
>
> [1] Zheng, Qinqing, et al. "Guided flows for generative modeling and decision making." arXiv preprint arXiv:2311.13443 (2023).
>
> [2] Zhang, Fan, and Michael Gienger. "Affordance-based robot manipulation with flow matching." arXiv preprint arXiv:2409.01083 (2024).
>
> [3] Liu, Xingchao, and Chengyue Gong. "Flow Straight and Fast: Learning to Generate and Transfer Data with Rectified Flow." The Eleventh International Conference on Learning Representations.

---

> > ### Comment · Reviewer_S8sj · 2025-08-07
> >
> > The authors have addressed my concerns adequately. Dear authors, please consider incorporating some of the information in  your rebuttal in your main paper as I believe they will be valuable addition to your work.

---

> > > ### Author Response · Authors · 2025-08-07
> > > **Reply to Official Comment by Reviewer S8sj**
> > >
> > > We appreciate your insightful comments and will be sure to incorporate the analysis on wall-time efficiency, information regarding the confidence interval and additional experiments in our revision.

---

### Official Review · Reviewer_6Jdc · 2025-07-01

**Clarity:** 2
**Significance:** 3
**Originality:** 3
**Rating:** 4
**Confidence:** 2

**Summary:**

This paper proposes ReinFlow, an online reinforcement learning framework for fine-tuning flow matching policies, especially those trained from imperfect imitation data. The method injects learnable, bounded Gaussian noise into the deterministic flow policy to turn it into a discrete-time Markov process. The paper demonstrates improvements across both locomotion and manipulation tasks (with state or pixel inputs, dense or sparse rewards), using as few as one denoising step, outperforming or matching baselines like DPPO and FQL while reducing wall-time by up to 80%.

**Questions:**

1. From Figure 6, it appears that the performance of ReinFlow is quite sensitive to small changes in noise magnitude and regularization strength, with no consistently clear trends. This raises the concern that significant per-task hyperparameter tuning may be required, which limits practical scalability. Do the authors have any thoughts on improving robustness or automatically adapting these hyperparameters during training? Also, could you clarify the meaning of $\alpha$ in this context?

Minor:
1. Figure 1 (c) and (d) should be "v3" instead of "v2"?
2. Missing caption for the last figure (figure 6).

**Ethical Concerns:**

["NO or VERY MINOR ethics concerns only"]

**Final Justification:**

Most of the concerns are addressed in the response, while the sensisitivity still remains an issue, the authors provided several possible solutions.

**Limitations:**

The authors have mentioned some future work in the last section.

**Quality:**

3

**Strengths And Weaknesses:**

Strength:
- First practical online RL framework for flow matching policies in robot learning. The proposed method is principled and mathematically sound.
- Good readability for most of the part.
- Significant wall-time savings compared to diffusion-based methods, and it works effectively with very few denoising steps, which is critical for real-time robot control.

Weakness:
- The authors claimed to "prioritize wall-time efficiency in simulated environments over sample cost", while they chose to present sample cost instead of wall-time efficiency for `Kitchen-mixed`, `Kitchen-partial` and `Robomimic` benchmarks. This creates a confusing inconsistency and makes it harder to verify their wall-time efficiency claims in those settings.
- Section 6 presents only partial results for the design choices and ablation studies. It's unclear whether the observed trends and results generalize across all tested environments.

---

> ### Author Rebuttal · Authors · 2025-07-29
>
> Thank you for the encouraging feedback!
>
> We appreciate your acknowledgement on the novelty and theoretical soundness of our algorithm, the clarity of our presentation, and ReinFlow's advantage in wall-time saving.
>
> We are happy to address your questions and concerns in what follows.
>
> ****
>
> **1. Use a Consistent Horizontal Axis**
> > "The authors claimed to 'prioritize wall-time efficiency in simulated environments over sample cost', while they chose to present sample cost instead of wall-time efficiency for Kitchen-mixed, Kitchen-partial and Robomimic benchmarks. This creates a confusing inconsistency and makes it harder to verify their wall-time efficiency claims in those settings."
>
> We thank the reviewer's helpful suggestion to emphasize wall-time efficiency in the main text. Due to space constraints, we included detailed wall-time statistics in Table 3 (page 28) of the appendix. In future revisions, we will follow your advice and include wall-time efficiency plots for the Franka Kitchen-mixed, Kitchen-partial, and Robomimic tasks directly in the main text to improve clarity and consistency of our work.
>
> ****
>
> **2. Ablation studies in More Environments**
>
> > "Section 6 presents only partial results for the design choices and ablation studies. It's unclear whether the observed trends and results generalize across all tested environments."
>
> Thank you for your feedback on the ablation studies. In Section 6, we presented 7 detailed sensitivity analyses across 5 representative environments, and we observed similar trends in other tasks.
> * During the rebuttal, we compare the effect of different types of noise conditions in Humanoid-v3 environment. We run the group that conditions only on the state ($\sigma(s)$) and the group that conditions on both state and time ($\sigma(s,t)$) with three random seeds.
> | Task    | $\sigma(s)$  | $\sigma(s,t)$  |
> |-------------------|--------------|------------------|
> | Humanoid| 4987.39	± 97.82  |  5076.12 ± 37.47   |
>
>   The results show that conditioning the noise on both time and state is beneficial to obtain higher reward, coinciding with the results in Fig. 5 (page 9).
>
> * We also train ReinFlow agent with different noise scales in Franka-kichen complete, and result is averaged over three seeds.
> | Noise std| Success Rate |
> |-------------|----------------------------------|
> | 0.001       | 70.42 % ± 3.21  %         |
> | 0.08        | 90.67 % ± 14.87 %         |
> | 0.16        | 99.08 % ± 1.01   %         |
>
>   This experiment reveals that a larger noise level promotes exploration and it is also supported by lines 262-265 (page 8).
>
> Due to time constraint in the rebuttal phase, we are still running the rest of the new experiments, and we will update results when we are ready and incorporate it in our subsequent revisions.
>
> ****
>
> **3. Understanding the Sensitivity Analysis**
>
> > "From Figure 6, it appears that the performance of ReinFlow is quite sensitive to small changes in noise magnitude and regularization strength, with no consistently clear trends. This raises the concern that significant per-task hyperparameter tuning may be required, which limits practical scalability. Do the authors have any thoughts on improving robustness or automatically adapting these hyperparameters during training? Also, could you clarify the meaning of $\alpha$ in this context?"
>
> We appreciate the reviewer’s insightful comments on hyperparameter sensitivity and offer the following clarifications:
>
> * **Trend in Noise Magnitude**.
>   - As discussed in lines 262 to 268 (page 8), small noise levels (0.001–0.040) leads to insufficient exploration, resulting in performance close to the pre-trained policy. However, a moderate noise range (0.04–0.20) yields high reward, with robustness to variations within this range. This demonstrates ReinFlow is **not overly sensitive** once past a noise threshold.
>   - Furthermore, we provide the chosen noise valuess for each task in the Appendix (page 33-35), where a noise range between 0.06 to 0.16 is reasonable to most tasks. Although it may yield some benefits, extensive per-task tuning is not necessary.
>
> * **Trend in Regularization Strength**
>
>   In Fig 6(a) (page 9), we illustrate the effects of two distinct types of regularization:
>   - **Wasserstein-2 Regularization** (blue curves, Eq. 9, page 6) with coefficient $\beta$ penalizes deviations from the pre-trained model. As expected, as $\beta$ increases from 0.01 to 1.0, the policy becomes more constrained, leading to a clear decline in reward back to the pre-trained level.
>    - **Entropy Regularization** (red curves, lines 195–205, page 6), with coefficient $\alpha$, encourages exploration and consistently achieves higher rewards. This suggests that entropy regularization is more effective than conservative constraints in our setting. The coefficient $\alpha$ is defined in Algorithm 1 (line 19, page 4).
>
>     To improve robustness, future work could explore **adaptive tuning** of $\alpha$ using primal-dual methods, as done in Soft Actor-Critic [1].
>
> ****
>
> **4. Typos**
> > "Figure 1 (c) and (d) should be  'v3' instead of 'v2'? "
> > "Missing caption for the last figure (figure 6)."
>
> We appreciate the reviewer's suggestions!. We will be sure to revise those typos based on your valuable feedback.
>
> ****
>
> **Final remark**
>
> We hope our response have addressed your concerns effectively, and we kindly ask whether you would consider improving our rating. We are happy to clarify any remaining questions. Thank you for your time and thoughtful feedback!
>
> ****
>
> **References**
>
> [1] Haarnoja, Tuomas, et al. "Soft actor-critic: Off-policy maximum entropy deep reinforcement learning with a stochastic actor." International conference on machine learning. Pmlr, 2018.

---

> > ### Comment · Reviewer_6Jdc · 2025-08-04
> >
> > Thanks for the detailed response and some additional experiments. These have largely addressed my concerns, and I am considering to raise my rating to weak accept.

---

> > > ### Author Response · Authors · 2025-08-04
> > > **Respond to Official Comment by Reviewer 6Jdc**
> > >
> > > Thank you for your consideration and kind response! We will follow your advice and strengthen the abalation studies and improve our presentation in our revisions.

---

### Official Review · Reviewer_h3PM · 2025-07-03

**Clarity:** 3
**Significance:** 3
**Originality:** 3
**Rating:** 4
**Confidence:** 4

**Summary:**

This paper proposes ReinFlow, which uses online reinforcement learning (RL) to fine-tune a family of flow-matching polices for continuous robotic control. The essential idea of ReinFlow is to inject noise into a flow policy’s deterministic path and convert the flow into a discrete-time Markov Process for tractable likelihood computation. This allows ReinFlow to stably fine-tune diverse flow models using online RL. Empirical evaluations on robotic locomotion and robotic manipulation tasks demonstrated that ReinFlow can increase success rate, episode reward, and save wall-time compared to the state-of-the-art diffusion RL method.

**Questions:**

1.	How does the noise injection net automatically balance exploration and exploitation?

2.	As claimed in the paper, the noise network enables ReinFlow to learn to create more diverse actions by altering the intensity of noise at different denoising steps. The entropy regularization in Section 4.4 also promotes agents to seek more diverse actions. If the noise network itself already enables ReinFlow to have a good exploration ability, what is the reason to introduce the additional entropy regularization? Does the entropy regularization further promote the exploration ability of agents? Is this promotion scalable to different tasks, instead of only in locomotion tasks?

**Minors:**

- Line 157: For completeness, we provide the proofs in Appendix ??.

**Ethical Concerns:**

["NO or VERY MINOR ethics concerns only"]

**Final Justification:**

The author's rebuttal has addressed my concerns. Thank you.

**Limitations:**

1.	The upper and lower bounds of the noise net is a hyper-parameter of ReinFlow that varies in different tasks and robot embodiments. This limits the scalability of ReinFlow to more diverse tasks without prior knowledge of the robot’s embodiments.

**Quality:**

3

**Strengths And Weaknesses:**

**Strengths**:

- This paper highlights the challenges of unstable back-propagation in fine-tuning conditional flow policies in very few steps for continuous control problems. In addition, designing an exploration mechanism for conditional flows with a deterministic path is also elusive. ReinFlow provides a stable fine-tuning solution, enables light implementation, and plugs into diverse flow models, e.g., ReFlow and Shortcut Models.

- The idea of converting flows to a discrete-time Markov Process using a noise injection network is interesting and efficient. ReinFlow uses RL to improve flow policies at very few or even one denoising step, and enables faster inference and high success rate after fine-tuning.
- The proposed method is evaluated in extensive experiments in robotic locomotion and robotic manipulation tasks. ReinFlow increases the success rate of pre-trained manipulation policies, and increases the episode rewards of locomotion policies significantly, with a reduction of wall-time compared the state-of-the-art diffusion RL method.

**Weaknesses**:

- Need more details and discussion on how the noise injection net can convert flow into a discrete-time Markov Process.

- Both noise injection and entropy regularization can enhance exploration. Missing discussion on the differences between the two methods.

- Hyperparameters potentially need to be tuned in different tasks.

---

> ### Author Rebuttal · Authors · 2025-07-29
>
> **General response to Reviewer h3PM**
>
> We thank Reviewer h3PM for the thoughtful review and encouraging comments. We appreciate your recognition of our well-motivated problem formulation, the flexibility of our noise injection technique, and the effectiveness and efficiency demonstrated in our extensive experiments.
>
> Below, we are happy to address your questions in detail.
>
> **1. Noise injection v.s. entropy regularization**
> > "Both noise injection and entropy regularization can enhance exploration. Missing discussion on the differences between the two methods. "
>
> > "As claimed in the paper, the noise network enables ReinFlow to learn to create more diverse actions by altering the intensity of noise at different denoising steps. The entropy regularization in Section 4.4 also promotes agents to seek more diverse actions. If the noise network itself already enables ReinFlow to have a good exploration ability, what is the reason to introduce the additional entropy regularization? Does the entropy regularization further promote the exploration ability of agents? Is this promotion scalable to different tasks, instead of only in locomotion tasks?"
>
> Thank you for raising this important point! Here's a clarification:
>
> * **The purpose of using noise and entropy**
> As described in Section 4.1 (page 4, lines 135–142), noise injection is primarily used to replace the $\delta(\cdot)$ function, making the action distribution tractable. The added stochasticity also naturally aids exploration. Entropy regularization (Section 4.4, page 6), on the other hand, provides an explicit mechanism to control exploration via its coefficient $\alpha$. Importantly, the method remains mathematically valid even without the entropy term.
>
> * **Interaction between noise and entropy.**
> We observe that during training, the noise network tends to reduce its output variance to stabilize returns, especially in tasks with dense rewards (e.g., locomotion). This behavior can prematurely dampen exploration, leading to convergence to suboptimal policies. Entropy regularization mitigates this by encouraging higher noise levels, thereby sustaining exploration throughout training.
>
> * **Entropy works in tasks beyond locomotion.**
>   - We successfully adopt entropy regularization in Robomimic visual manipulation task "square", as is shown in Table 9 b, page 35 in the Appendix.
>   - This technique also yields additional success rate in "Franka Kitchen-complete", which is a state-input, sparse-reward manipulation task. This is supported by an ablation experiment conducted during the rebuttal, where results are averaged over 3 seeds:
> | Without Entropy Reg.  | With Entropy Reg. ($\alpha=0.01$) |
> |-------------------------------|-----------------------------------------|
> | 96.17 ± 3.65%             | 99.00 ± 0.75%                         |
>
> We will follow your insightful advice and add a dedicated discussion to clarify the relationship between noise and entropy in the revised manuscript.
>
> **2. Conversion to a Discrete-time Markov Process**
> > "Need more details and discussion on how the noise injection net can convert flow into a discrete-time Markov Process."
>
> Thank you for the suggestion on emphasizing the functionality of the noise injection net. We explained in Section 4.1(lines 135 to 143, page 4) that the noise conditions on the current state and action, ensuring that the resulting process maintains the Markov property. We will follow your advice and highlight this mechanism in our revision.
>
> **3. Automatic Exploration–Exploitation Trade-off**
> > "How does the noise injection net automatically balance exploration and exploitation?"
>
> That is an insightful observation! As described in Section 6 (page 8), the stochasticity introduced by the noise net promotes exploration early in training, allowing the policy to exceed pre-trained performance. Over time, we also observe the network learns to reduce noise levels, shifting toward exploitation. We will emphasize this adaptive behavior in the revised manuscript.
>
> **4. Tuning the hyperparameter of noise levels.**
> > "The upper and lower bounds of the noise net is a hyper-parameter of ReinFlow that varies in different tasks and robot embodiments. This limits the scalability of ReinFlow to more diverse tasks without prior knowledge of the robot’s embodiments."
>
> Thank you for your thoughtful review! We report the noise bounds in Tables 7 to 9 (page 33 to 35 in the Appendix). While task-specific tuning may yield minor gains, we find (lines 262 to 265, page 8) that ReinFlow is relatively robust to noise level changes within a reasonable range: a noise standard deviation of 0.08-0.12 generally works well across most tasks. Nonetheless, reducing ReinFlow's sensitivity to noise level is a promising direction for future work!
>
> **5. Typo**
> > "Line 157: For completeness, we provide the proofs in Appendix ??."
>
> Thank you—the proof presents in Appendix B (page 22). We will correct this and ensure all appendix references are properly linked.
>
> **Concluding remark**
>
> We hope our response has addressed your concerns and the questions raised. We would be grateful if you kindly consider improving our rating. Please do not hesitate to let us know if you have any further questions.

---

### Note · Authors · 2025-08-15

We thank the reviewers for their detailed and thoughtful feedback, as well as their patience and advice during the rebuttal phase.

**Summary of the rebuttal phase**

- **Reviewer h3PM**

  - We clarify the distinction between noise and exploration and present Franka Kitchen experiments showing the applicability of entropy regularization beyond locomotion.

  - We describe how noise injection constructs a discrete-time Markov process and how learnable noise automatically balances exploration and exploitation.

  - We highlight our method is relatively robust to hyperparameters within a reasonable range.

- **Reviewer 6Jdc**

  - We conducted additional ablation studies in Humanoid-v3 and Franka Kitchen-complete to show that our conclusions regarding key hyperparameters remain consistent across diverse tasks.

  - For the sensitivity analysis, we analyze the observed trends in noise magnitude and regularization, which will be emphasized in the revised manuscript. We will also ensure a consistent horizontal axis in the revised figures.

- **Reviewer S8sj**

  - We explain ReinFlow’s significant wall-time reduction with new experiments in Hopper, Walker, and Kitchen-complete, which show that our speed advantages arise from both faster sampling and more efficient optimization. We will add a separate section discussing this in our revision.

  - We also appreciate the reviewer’s suggestions on formatting improvements, emphasizing confidence intervals, and including additional experiments for state-input tasks, and we plan to incorporate these changes.

- **Reviewer aYCb**

  - We clarify ReinFlow and SAC are designed for very different purposes, ReinFlow excels in complex manipulation tasks where SAC typically fails. We emphasize that ReinFlow differs from DPPO by focusing on flow policies with an ODE path, which is gaining traction in robot learning community.

  - We explain why ReinFlow is faster than DPPO in principle, our reason for not using a Monte Carlo estimate of the log probability, and ReinFlow’s pioneering role in training flow policies for robotics.

  - We conducted additional experiments across all tasks using five new random seeds, which show minimal deviation from the original results, confirming ReinFlow’s robustness.

  We appreciate reviewer aYCb’s suggestions regarding presentation, comparison with SAC/DPPO, speed, and reproducibility. We will make sure to address these points in the revision.

---

### Decision · Program_Chairs · 2025-09-17

**Decision:**

Accept (poster)

**Comment:**

The paper proposes a new approach to perform online finetuning of flow-matching-based policies through a specific way to inject noise in the diffusion process, which converts the flow formulation into a discrete-time Markov chain with tractable likelihood computation. The empirical shows that the resulting method can significantly improve the initial policy in simulated locomotion and manipulation domains while reducing the computational complexity wrt to flow-based baselines.

In the rebuttal phase, the authors addressed most of the concerns from the reviewers, notably expanding the empirical validation to include additional ablations (e.g., to clarify the difference between noise injection and entropy regularization) and clarified aspects such as wall-time computation and comparison with baselines. One remaining among reviewers after the rebuttal was the marginal improvement obtained wrt to SAC and DPPO (at least in some domains). While the authors pointed out that SAC and ReinFlow are effective in different domains and DPPO is less efficient, I think this stands as a valid concern and I strongly encourage the authors to acknowledge this aspect in the updated version of the paper and expanding on the discussion in the rebuttal. Despite this aspect, there is an increasing evidence that diffusion/flow-based policies are the most effective representation in many robotic domains and proposing a computationally efficient and performance effective fine-tuning mechanism is of paramount importance in this context. For this reason and given the overall consensus from reviewers, I propose acceptance for the paper.